# mCLM: A Modular Chemical Language Model that Generates Functional and Makeable Molecules

**Carl Edwards**[1*]     **Chi Han**[1*]     **Gawon Lee**[2]     **Thao Nguyen**[1]     **Sara Szymkuć**[3]

**Chetan Kumar Prasad**[2]     **Bowen Jin**[1]     **Jiawei Han**[1]     **Ying Diao**[4]     **Ge Liu**[1]

**Hao Peng**[1]     **Bartosz A. Grzybowski**[5,6]     **Martin D. Burke**[2]     **Heng Ji**[1]

## Abstract

Despite their ability to understand chemical knowledge, large language models (LLMs) remain limited in their capacity to propose novel molecules with desired functions (e.g., drug-like properties). In addition, the molecules that LLMs propose can often be challenging to make, and are almost never compatible with automated synthesis approaches. To better enable the discovery of functional small molecules, LLMs need to learn a new molecular language that is more effective in predicting properties and inherently synced with automated synthesis technology. Current molecule LLMs are limited by representing molecules based on atoms. In this paper, we argue that just like tokenizing texts into meaning-bearing (sub-)word tokens instead of characters, molecules should be tokenized at the level of functional building blocks, i.e., parts of molecules that bring unique functions and serve as effective building blocks for real-world automated laboratory synthesis. This motivates us to propose mCLM, a modular Chemical-Language Model that comprises a bilingual language model that understands both natural language descriptions of functions and molecular blocks. mCLM front-loads synthesizability considerations while improving the predicted functions of molecules in a principled manner. Experiments on 430 FDA-approved drugs showed that mCLM is capable of significantly improving chemical functions critical to determining drug potentials. mCLM, with only 3B parameters, also achieves improvements in synthetic accessibility relative to 7 other leading generative AI methods including GPT-5. When tested on 122 out-of-distribution medicines using only building blocks/tokens that are compatible with automated modular synthesis, mCLM outperforms all baselines in property scores and synthetic accessibility. mCLM can also reason on multiple functions and iteratively self-improve to rescue drug candidates that failed late in clinical trials ("fallen angels").

🤗 Data and Model    ⌨ Code

## 1 Introduction

Small molecules—the class of chemical matter primarily built from carbon atoms bonded together—can perform a wide range of important functions in human society (Zhang et al., 2025). These include essentials like promoting health by acting as medicines (Zheng et al., 2024; Edwards et al., 2024b; Singhal et al., 2023; Thirunavukarasu et al., 2023; Xiao et al., 2024a), converting energy by functioning as key components in solar cells (Nguyen et al., 2024b; Lv et al., 2021; Li et al., 2023b; 2024b; Si et al., 2024), and achieving sustainability by serving as inherently recyclable products. These functions also include many nice-to-haves that drive substantial economic growth, including colorants, flavorings, perfumes, cosmetics, coatings, quantum dots and insect repellants.

---

*Equal contribution. [1]Siebel School of Computing and Data Science, University of Illinois Urbana-Champaign, [2]Department of Chemistry, University of Illinois Urbana-Champaign, [3]Allchemy Inc., [4]Department of Chemical and Biomolecular Engineering, University of Illinois Urbana-Champaign, [5]Ulsan National Institute of Science and Technology, [6]Institute of Organic Chemistry, Polish Academy of Sciences. Correspondence: cne2@illinois.edu, chihan3@illinois.edu, hengji@illinois.edu, mdburke@illinois.edu

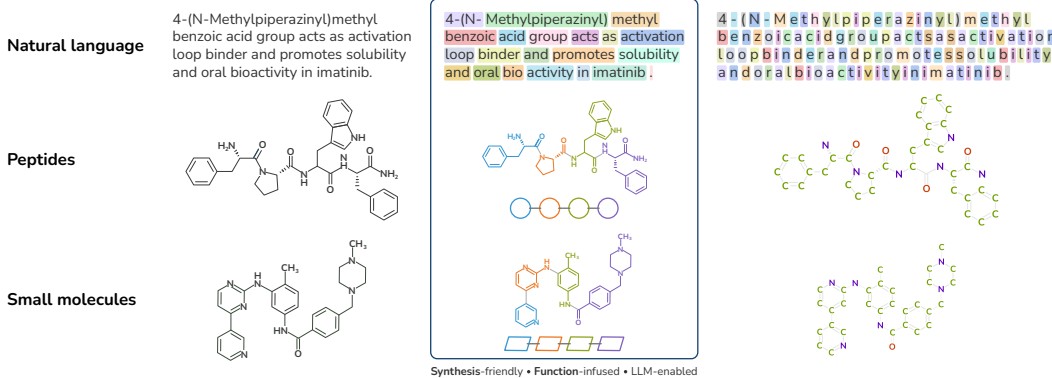

Figure 1: mCLM adopts a modular chemical vocabulary, which uses synthesis robot-friendly molecular building blocks as tokens together with natural language tokens. Compared with using natural language names, SMILES strings, or holistic embeddings for whole molecules, this building block level of tokenization better enables the prediction of compounds with improved properties, and guarantees automated synthesizability a priori, thus building a direct link between the digital and physical worlds. This approach stands to substantially enable AI-guided discovery of new small molecules with targeted functions.

The traditional approach for small molecule synthesis is highly artisanal, and thus slow and expensive: (1) It is unfriendly to automation: machines are not good at performing thousands of reaction types, with each run under thousands of possible conditions and using millions of possible starting materials. (2) It leads to an undemocratized landscape in drug discovery. With development costs averaging around $1.3 billion per drug, only economically advanced countries can afford to invest in such high-risk research (Kneller, 2010). It also excludes potential breakthroughs from the developing world and inhibits research that could prevent diseases prevalent there. Moreover, participation in the process of molecular innovation currently requires access to highly trained experts in chemical synthesis. As a result, there are no effective drugs for many diseases such as Parkinson and Amyotrophic Lateral Sclerosis. (3) Most importantly, there are many instances of known commercial drugs or materials that have well-documented limitations that have remained unaddressed. For example, Imatinib is an important anticancer drug that works well in most parts of the body, but it only poorly penetrates the blood-brain barrier (Takayama et al., 2002; Senior, 2003; Coutre et al., 2004; Isobe et al., 2009). This means it may effectively treat cancer at its primary site, yet fail to prevent fatal brain metastasis. In such cases, more blood-brain-barrier penetrant molecules that retain all of the other favorable properties of the current drug are desired, but making such selective modifications in functional properties can be very challenging. As another example in the materials domain, 350+ million people in Asia and the Pacific have only limited access to electricity, and 150 million people still have no access at all. Organic photovoltaic (OPV) molecules are expected to be more economical and environment-friendly alternatives to current solar cells. However, current commercialized OPV devices either achieve less than 10% energy conversion efficiency—significantly lower than traditional silicon solar cells—or have stability far short of 10 years (Solak and Irmak, 2023).

An alternative block chemistry approach for small molecule synthesis has recently emerged (Gillis and Burke, 2007; Woerly et al., 2014; Li et al., 2015a; Lehmann et al., 2018; Trobe and Burke, 2018; Blair et al., 2022a; Angello et al., 2022; Wang et al., 2024; Strieth-Kalthoff et al., 2024a). Block chemistry is iterative carbon-carbon bond-forming chemistry that machines can do. It iteratively assembles small molecules from prefabricated building blocks using chemistry that is simple and general and thus readily automated. Akin to automated DNA, RNA, and peptide synthesis platforms, a major strength of this block-based approach is that it can access billions of novel small molecules with high degrees of functional potential using only a few automation-friendly reactions and a bounded set of pre-fabricated function-infused building blocks. We specifically recognized that this block-based approach provided an opportunity to create a new modular language for chemistry. The idea that molecules and their synthesis may be best understood from the perspective of a "chemical language" dates back to 2014 (Cadeddu et al., 2014), where structural fragments and functional groups play the roles of "chemical words". This view is supported by multiple observations aligned with natural language: (1) they can be decomposed and reassembled, (2) they exhibit ambiguity—the

same building block can perform vastly different functions depending on the chemical context, and (3) they possess significant diversity, as many different structures can lead to the same function. This linguistic parallel suggests the potential to train a large language model specifically for molecules by linearizing their structures into modular sequences.

A common representation for such an approach is the Simplified Molecular Input Line Entry System (SMILES; Weininger, 1988), where atoms are denoted by one- or two-character symbols (e.g., C for carbon, Br for bromine, and F for fluorine), rings are represented with numbers, and branches are indicated using parentheses. However, such atom-level tokenization strategies (e.g., SMILES (Weininger, 1988) or SELFIES (Krenn et al., 2020)) resemble character-level natural language models, which struggle to generalize effectively. Unlike proteins, which have a fixed vocabulary of 20 amino acids, small molecules exhibit an open vocabulary when each atom is treated as a token, as illustrated in Figure 1. It also causes severe restrictions in practicality because many of the new structures proposed by LLMs based on atom-level tokenization are not practically synthesizable in the laboratory. This approach has created a major gap between what is now possible in silico and what is possible in the physical world.

Furthermore, the SMILES representation can obscure critical structure information: two atoms that are direct neighbors in the molecular graph may be distantly separated in the SMILES string. Even with recent efforts to align SMILES and natural language description (Edwards et al., 2022a; Pei et al., 2024; Ahmad et al., 2022), integrate chemical properties and functional groups (Nguyen et al., 2025) into SMILES, incorporate 3D geometric information (Fu et al., 2025; Li et al., 2025a), and employ graph neural networks to capture molecular graphs and chemical reaction contexts (Wang et al., 2022a), these representations still fail to encapsulate functional knowledge that is often described only in natural language literature because the inherent properties and functions of molecules are hidden in their structure, composition, and interaction.

Therefore, our goal is not to make all drug-like molecules; rather, our goal is to figure out how to make the right ones, better, faster, stronger. Unlike machines, human scientists are inherently "multilingual," seamlessly navigating diverse modalities—from natural language and scientific figures in literature to complex scientific data such as molecular structures in knowledge bases. In contrast, most prior work on molecule discovery trains large language models on a single modality. Moreover, Human scientists "think before they talk," grounding their reasoning in deeply reflective and deliberate reflection and critical evaluation to generate new hypotheses. Current models lack this critical thinking capacity, limiting their ability to contribute meaningfully to discovery. In particular, the human body is a highly complex, interconnected system, and drug discovery is essentially a multi-objective optimization problem. It involves balancing factors such as drug absorption, first-pass metabolism, bioavailability, distribution, protein binding, and clearance. However, improving one property often comes at the expense of another. Many promising drug candidates have failed in the final stages of FDA approval due to an unacceptably high risk of drug-induced liver injury.

Against this backdrop, we argue that there is a correctable fundamental mismatch between the way LLMs work and the way chemists traditionally synthesize and study small molecules. Reducing tokenization granularity to the level of individual letters is counterproductive, as it complicates meaning extraction and increases the likelihood of generative AI hallucinating words that don't exist. Analogous limitations are inherently linked to atom-based tokenization. And as a result, unfortunately, much of the generative AI research for scientific discovery within the computer science community does not extend to hypothesis verification in the physical world. To bridge this gap, in this paper, we propose a novel LLM by drawing inspiration from the scientific discovery process itself. We aim to develop a science-inspired large language model that follow three principles: (1) "Observe" - acquire, represent and integrate knowledge from multiple data modalities; (2) "Think" – think critically to generate hypotheses; and (3) "Propose" – verify hypotheses through the Physical World. We aim to teach computers to speak two complementary languages: one that represents molecular building blocks (i.e., subgraph structures) indicative of specific functions and compatible with automated modular assembly, and another that describes these functions in natural language (Figure 2). Unlike existing approaches that add such knowledge as a post hoc step, we develop a function- and synthesis-aware modular chemical language model (mCLM). Inspired by bilingual speakers who frequently "code-switch" (naturally and often switch between their two languages within the same message; Poplack, 2013), we propose a novel neural encoder that integrates molecular structure and natural language. mCLM incorporates both function- and synthesis-related knowledge into the small molecule tokenization process a priori. First, we tokenize small molecules at the level

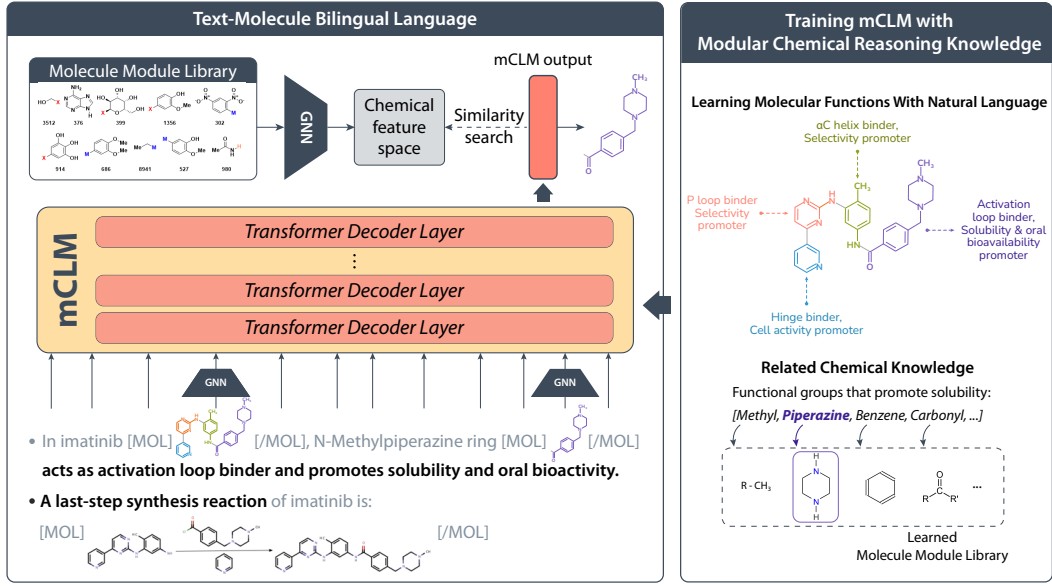

Figure 2: An overview of the mCLM. It is a multimodal chemical-language model that tokenizes molecules into synthesis robot-friendly *building blocks*, thus creating a direct link between the digital and physical worlds. After being trained on datasets consisting of properties, functions and synthesis data, the mCLM can conduct critical chemical reasoning through an iterative refinement process.

of building blocks (graph substructures) that are able to predict function and are, by design, flexible, multi-scale, fully compatible with automated modular small molecule synthesis. We use graph neural networks to encode each building block. We then extract natural language sentences from the literature that describe the molecule's functions and chemical reactions and synthesis constraints of various molecules, and seamlessly insert the encoded building blocks alongside the corresponding entity names to form the training data, as illustrated in Figure 2. By reasoning on such functional building blocks, mCLM guarantees to generate efficiently synthesizable molecules thanks to recent progress in block-based chemistry, while also improving the functions of molecules in a principled manner. During model inference, mCLM enables these functional modules to be predictably and automatically assembled into molecules with desired functions.

Compared to previous work, mCLM has multiple potential advantages of encoding and generating molecules at a modular level:

1. **Synthesis efficiency**: proposed molecules can be faster and more broadly makeable because the process is, by design, simple, iterative, general, and machine-friendly. This can enable rapid iterative drug and material development via seamless integration with automated lab experiments.
2. **Multimodal understanding**: resembling the mechanism of natural language tokenization, molecular modularization provides a more natural interface to align with word representations to form a more powerful multimodal representation.
3. **Deliberate reasoning**: leveraging LLMs' instruction following capability and wide-scale pretraining, mCLM is able to iteratively refine molecules based on knowledge of molecular functions.

## 2 THE MODULAR CHEMICAL LANGUAGE MODEL

In natural language modeling, tokenization identifies common substrings (such as words and sub-words), which carry richer semantic information than sequences of characters. Similarly in nature, most small molecules are composed primarily of connected *building blocks* (Lehmann et al., 2018; Trobe and Burke, 2018). There is likewise a high degree of inherent modularity in many medicines and materials (Ertl and Schuhmann, 2019; Arkan et al., 2020; Andrews et al., 1984; Vitaku et al., 2014). To leverage this phenomenon in chemistry and borrow the spirit from natural language modeling, we propose mCLM, a multimodal model that jointly encodes and understands natural language and molecules based on synthesis-friendly building blocks instead of atoms. In Section 2.1,

we introduce the concept of molecular building blocks as a chemical "vocabulary" and describe the tokenization process to obtain the library of building blocks. Then in Section 2.2 we describe the mCLM architecture and its training. Finally, we introduce the reasoning mechanism of mCLM which refines molecule design over multiple iterations in Section 2.3.

## 2.1 A FUNCTION-INFUSED AND SYNTHESIS-FRIENDLY VOCABULARY

In this work, we propose to leverage a chemical vocabulary $V$ of synthesis-friendly building blocks. This approach guarantees capacity for automated iterative assembly first reported in 2015 (Li et al., 2015b) and, since then, demonstrated in multiple experimental campaigns including Strieth-Kalthoff et al. (2024b) and Angello et al. (2024). The blocks can be chemically connected using predefined synthesis rules in a short period. Briefly, akin to language models developed for peptides/proteins, small molecules can be assembled automatically from makeable building blocks. These building blocks are often highly associated with chemical functions, such as binding to protein targets, modulating enzyme activity, or affecting involved metabolic processes. This will enable rapid and iterative proposal of new small molecules, automated synthesis of those small molecules, and generation of the corresponding functional data on demand. In contrast to SMILES strings which break molecule structure during graph linearization (e.g., separating physically adjacent subgraphs such as the two carbons in molecule C(N)C), our representation is linear in the physical world.

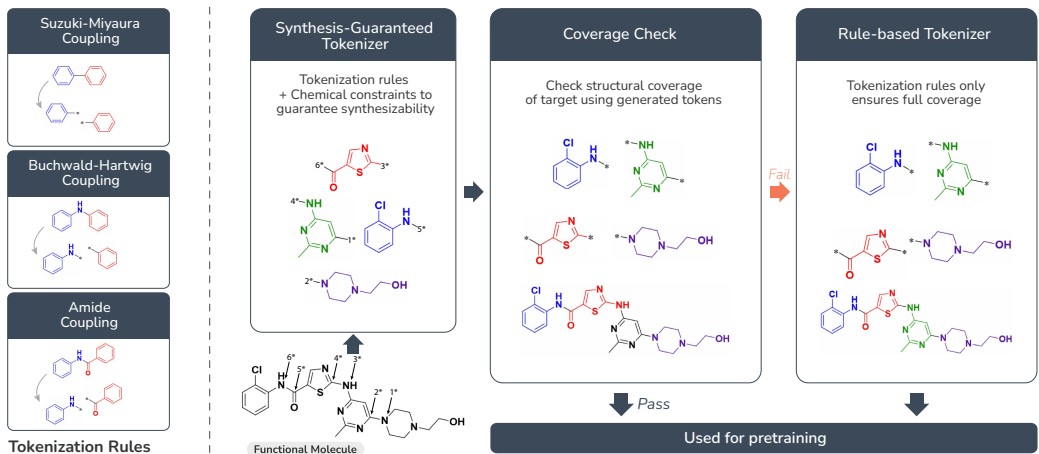

Figure 3: An overview of the tokenization process. A functional molecule is first processed by the synthesis-guaranteed tokenizer to produce a set of building blocks compatible with automated modular synthesis. These blocks are then evaluated via a structure coverage check to determine whether they fully reconstruct the original molecule. If coverage is complete, the blocks are used directly for pretraining. Otherwise, the molecule is reprocessed using a rule-based tokenizer to ensure full representation for training purposes.

During dataset construction, we combine two tokenization strategies to balance coverage and synthetic feasibility: a synthesis-guaranteed tokenizer and a rule-based tokenizer (Figure 3). The synthesis-guaranteed tokenizer disconnects the molecule only at bonds that can be formed by a predetermined small set of reactions that can be performed in an automated manner. Specifically, it only permits bond disconnections that correspond to reactions compatible with state-of-the-art automated synthesis platforms. These three bonds are: amide coupling, Suzuki-Miyaura coupling, and Buchwald-Hartwig coupling (see Figure 3 tokenization rules; Tyrikos-Ergas et al., 2025). The disconnection rules are adapted from well-established principles in block-based and automated chemistry (Blair et al., 2022b; Li et al., 2015b). Qualitative examples of tokenization are included in the appendix F, and the full rules and tokenizer code will be released with documentation. Importantly, the tokenizer is modular: as new automated reactions are validated in practice, their corresponding disconnection rules can be readily incorporated, expanding the makeable chemical space without requiring fundamental architectural changes. Unlike traditional retrosynthesis tools that prioritize maximum coverage or human-designed heuristics, this tokenizer is guided by the operational constraints of block chemistry. It also ensures that resulting blocks are free from functional group conflicts that would interfere with downstream synthesis. This conservative approach leads to a set of molecular tokens that are

guaranteed to work under defined synthesis protocols, enabling seamless transition from model-generated output to real-world synthesis. When the synthesis-guaranteed tokenizer cannot fully cover a molecule—due to chemical incompatibilities or reaction constraints—we fall back on a rule-based tokenizer to ensure coverage of a larger variety of molecules. The rule-based tokenizer also breaks molecules along the same automated machine-friendly bonds as the above tokenizer. However, no other rules are applied beyond specifying a minimum size for blocks. This rule-based tokenizer is used only during training to support learning on diverse molecular structures while maintaining consistency with synthesis logic. We note that the vocabulary growth is analogous to the shift from character- level to subword-level tokenization in natural language, where larger but semantically meaningful tokens significantly improved performance. These reusable, synthesis-friendly building blocks are derived from real data distributions. We collected approximately 8 million molecules from a wide range of public sources (Appendix G), of which approximately 6.8 million were tokenizable using our full tokenization pipeline (see Figure 3). This process resulted in a vocabulary of 800,000 unique building blocks with 200,000 synthesis-guaranteed (see Table 6). An example of how the block sequence can reconstruct the molecule structure is also provided in Appendix F.

## 2.2 CHEMICAL-LANGUAGE MODELING

Figure 2 illustrates the architecture of mCLM. The model is a sequential generative model that processes molecule and text sequences in a unified manner. It adopts a Transformer architecture, which is well-suited for handling sequential data and allows for using pre-trained language models as a backbone. After tokenizing the molecules as mentioned in the last section, we encode each building block using graph neural networks (GNNs; Edwards et al., 2024a; Sprueill et al., 2024; Gasteiger et al., 2021) before passing them through an adaptor module. These representations are then concatenated with natural language embeddings at positions where molecule entity names appear. This results in a form of "code-switched" language which blends molecular structures with natural language descriptions. The feature sequence is then fed into a Transformer decoder-only architecture, which predicts the next token based on previous tokens. This allows for pre-training the model on a large corpus of multi-modal data, enabling it to learn and "talk about" the relationships between the different modalities and their respective representations.

We train mCLM on top of open-source pretrained large language models (we adopt Qwen2.5 Yang et al. (2024) as the starting point), allowing us to leverage their natural language understanding and reasoning capabilities without incurring the computational cost of training from scratch. Furthermore, pretrained natural-language LLMs have been exposed to a broad range of scientific information, making them a strong foundation for our curriculum. They help the model acquire general scientific knowledge before progressing to domain-specific training on molecular structures and drug-related information. As the training objective, we adopt a unified categorical cross-entropy (CCE) loss applied to both natural language and molecular tokens:

$$\mathcal{L} = H(P(\mathbf{x}), P_\theta(\mathbf{x})), \quad \text{logit}\left(P_\theta(v \mid \mathbf{x}_{1\cdots i-1})\right) = \begin{cases} \mathbf{c}_i^\top \mathbf{e}_v, \ v \in \mathcal{V}_{\text{natural language}} \\ \mathbf{c}_i^\top f_\psi\left(\text{GNN}_\phi(v)\right), \ v \in \mathcal{V}_{\text{molecular building block}} \end{cases}$$

Specifically, the cross-entropy loss is computed between the ground truth distribution $P(\mathbf{x})$ and the model distribution $P_\theta(\mathbf{x})$ over the combined vocabulary of natural language and molecular building blocks. The model generates logits for the next token $v$ by computing the dot product between the contextual representation $\mathbf{c}_i$ with token embeddings. $\mathbf{c}_i$ is produced by the Transformer given the previous tokens $\mathbf{x}_{1\cdots i-1} = [X_1, \cdots, X_{i-1}])$. For natural language tokens, the embedding $\mathbf{e}_v$ is directly taken from the pretrained natural language model. The embedding for molecular building blocks is computed by passing the building block's graph through a GNN, followed by a linear adapter function $f_\psi$ to project it into the same embedding space. This formulation enables joint training over both modalities using a single loss function. We train on approximately 1 million examples consisting of the 1,000 most frequent molecular building blocks, sampled from a new dataset we constructed. This dataset contains over 36 million molecular instructions, 1,000 tasks, and 8 million molecules. More details on the training procedure are described in Appendix E. Notably, the architecture of mCLM is compatible with larger vocabularies, and we anticipate future scaling studies as computational resources permit. mCLM is able to generalize to molecules composed of unseen building blocks. During training, the model learns to associate chemical structure with function via instruction-based pretraining. During inference, new blocks can be directly incorporated without retraining.

## 2.3 CRITICAL CHEMICAL REASONING

Chemical reasoning over molecules often involves optimizing multiple functions, such as toxicity, bioactivity, and binding affinity. Therefore, it is not a straightforward task to propose an ideal molecule structure, especially with only a single attempt. Optimizing one function may lead to trade-offs in others. For example, many drugs with higher potency were rejected due to increased toxicity to patients. To address this, we propose a reasoning process that allows the model to refine its own generated molecules and iteratively improve their desired functions. At each iteration, we evaluate the properties and identify one that still requires improvement, and mCLM proposes a modification of the molecule targeting this property. This process is repeated until a maximum number of iterations is reached. This process is summarized in Appendix Algorithm 1.

## 3 EXPERIMENTAL EVALUATION

### 3.1 CREATING ORACLE MODELS FOR EVALUATION

To evaluate the performance of our generated molecules, we construct oracle models focused on Absorption, Distribution, Metabolism, Excretion, Toxicity (ADMET) property prediction. We select 6 tasks from the Therapeutics Data Commons (TDC) benchmark (Huang et al., 2021): **AMES** (mutagenicity), **BBBP** (blood-brain barrier permeability), **CYP3A4** inhibition (metabolism), **DILI** (drug-induced liver injury), **HIA** (human intestinal absorption), and **PGP** (P-glycoprotein substrate classification). The detailed training procedure is described in Appendix D.2.3.

### 3.2 IMPROVING FDA-APPROVED DRUGS WITH OUT-OF-VOCABULARY BLOCKS

In practice, real-world drug discovery is often driven by the optimization of known molecules, as existing approved drugs offer a more direct and promising path to safe and effective new drugs. Therefore, we evaluate mCLM's drug proposal capability in improving FDA-approved drugs. Specifically, we apply the mCLM to improve the 6 properties for all FDA-approved drugs consisting of synthesis-guaranteed blocks. This amounts to 122 molecular structures and 153 unseen blocks. We confine the output vocabulary of mCLM to 582 synthesis-guaranteed blocks. Most of these drugs (120/122) contain blocks that were not present in the 1,000-block training vocabulary, presenting an opportunity to examine how the mCLM works on out-of-distribution molecules. Results in Table 1 show that improvements are achieved for all properties[*]. This shows that mCLM is not limited to a fixed vocabulary but can generalize beyond its training coverage. We also compare our approach against a wide range of strong text-based molecule editing baselines, including MoleculeSTM (Liu et al., 2022), FineMolTex (Li et al., 2025b), GPT-4o, GPT-5, Gemini-2.5-Flash (Gemini-2.5-F), LDMol (Chang and Ye, 2024), and Claude 3.5 Haiku (Claude-3.5-H). We conducted additional comparisons with two relevant baselines: DGAE (Boget et al., 2024) (a discrete graph autoencoder using vector quantization) and HierVAE (Jin et al., 2020) (a hierarchical VAE for molecular graph generation). The detailed results and a technical comparison with vector-quantized methods are enclosed in Appendix H. Despite the fact that *all* of the baseline models are allowed to generate without synthesis guarantees, their average improvements fall behind mCLM.

Table 1: Average pharmacokinetic and toxicity properties of FDA drugs composed of synthesis-guaranteed blocks, as well as their proposed modifications. (↓: lower is better, ↑: higher is better). Green = better than FDA, Red = worse, Light green bold = best overall per column.

| Model | AMES (↓) | BBBP (↑) | CYP3A4 (↓) | DILI (↓) | HIA (↑) | PGP (↓) | Avg. Improv. |
|---|---|---|---|---|---|---|---|
| **FDA Drug** | 47.8 | 61.4 | 2.1 | 60.1 | 98.96 | 64.6 | 0.00 % |
| **MoleculeSTM** | 47.1 | 63.4 | 2.2 | 59.3 | 98.76 | 64.1 | 0.31 % |
| **FineMolTex** | 47.5 | 66.0 | 2.4 | 59.5 | 98.84 | **64.0** | -0.73 % |
| **LDMol** | 49.0 | 63.5 | 2.5 | 57.5 | 99.0 | 67.6 | -3.07 % |
| **GPT-4o** | 46.0 | 72.1 | 2.2 | 60.8 | **99.3** | 65.2 | 2.45 % |
| **GPT-5** | 49.1 | 70.2 | 2.4 | 61.2 | 99.2 | 65.0 | -0.82 % |
| **Claude-3.5-Haiku** | 49.3 | 72.2 | 2.2 | 58.6 | 98.3 | 65.7 | 1.64 % |
| **Gemini-2.5-Flash** | 45.0 | 72.5 | 1.8 | 58.1 | 99.1 | 64.5 | 6.98 % |
| **mCLM (Ours)** | **44.4** | **85.2** | **1.4** | **53.7** | 98.99 | 64.4 | **15.0 %** |

The key to expediting the drug creation process is to discover potent molecular candidates that are simultaneously synthesis-friendly. While mCLM shows strong property editing results, its key benefit

---

[*]We also test a different distribution of 430 non-synthesis-guaranteed FDA drugs in Appendix A.2.

lies in its synthesis-friendly nature. We assessed the synthesizability of generated molecules by computing synthetic accessibility (SA scores) (Ertl and Schuffenhauer, 2009) as a quick heuristic (see Table 3). Then, as a more rigorous assessment, we consider Allchemy, which is the state-of-the-art retrosynthesis software (Wołos et al., 2022; Strieth-Kalthoff et al., 2024c). Allchemy is computationally expensive, but it evaluates synthesizability to the best of publicly available human chemical knowledge. For example, it finds synthetic routes for 98.1% of the FDA-approved molecules. Moreover, to our knowledge it is the only retrosynthesis software for which many of the proposed routes have been reduced to practice in the lab with physical experimentation. These studies have been published in top tier journals (Mikulak-Klucznik et al., 2020). We select the top 3 natural language models by SA score (mCLM, MolSTM, and FineMolTex) and randomly sample 200 generated molecules from each to be assessed by Allchemy on a supercomputing cluster.

Table 2: Synthetic accessibility (SA) (Ertl et al., 2009), validity, and retrosynthetic results across baselines. Synthesizability is the percent of valid molecules where a retrosynthetic route was found. Makeability is the overall percent of generations which can be synthesized (Makeability =Valid $\times$ Synth.).

| Model | SA ($\downarrow$) | Validity (%) | Synthesizability (%) | Makeability (%) |
|---|---|---|---|---|
| FDA | 2.70 | **100.0** | 98.11 | 98.11 |
| MoleculeSTM | 2.64 | 93.80 | 91.03 | 85.39 |
| FineMolTex | 2.58 | 94.20 | 90.15 | 84.96 |
| mCLM (Ours) | 2.43 | **100.0** | **98.23** | **98.23** |
| mCLM (No GNN) | 2.97 | 45.21 | 94.66 | 42.8 |
| mCLM (No Synth. Tokenizer) | 3.09 | **100.0** | 73.65 | 73.65 |

As shown in Table 2, molecules proposed by mCLM are 100% valid (syntactically correct) and 98.2% synthesizable, which is superior even to the FDA drugs. In contrast, MoleculeSTM outputs are valid only 93.9% of the time, and among those, only 90.3% are predicted to be synthesizable using exhaustive retrosynthesis search. Out of MoleculeSTM-generated molecules, only 84.8% can be made ($93.9 \times 0.903 = 84.8$).

### 3.3 CASE STUDY: MULTI-STEP REASONING TO RESURRECT THE "FALLEN ANGELS"

There are many new drug candidates that almost reach FDA approval but fall short for various reasons when tested in clinical trials. For example, Evobrutinib is a Bruton's tyrosine kinase (BTK) inhibitor that went through clinical trial as a drug for relapsing Multiple Sclerosis. However, the FDA placed a partial clinical hold on Phase III trials in April 2023 after two patients showed signs of drug-induced liver injury (Montalban et al., 2024). TNG348 is a USP1 inhibitor designed for treating BRCA1/2-mutant and HRD cancers, but it failed in phase 1/2 clinical trials due to liver abnormalities (Inc, 2024; Simoneau et al., 2025). These "fallen angels" represent tremendous opportunities for impactful engagement of the AI/chemistry interface, because much is known about the strengths of each of these small molecules, and it is also known why they fell short. Fixing such fallen angels is a high leverage opportunity for the function-infused mCLM to contribute.

Figure 4 shows an application of the mCLM, without synthesis restrictions on the vocabulary, to these two molecules. For both, the initial step is to optimize DILI, the reason the drugs failed in clinical trials. Following that, the mCLM fixes other properties which were made worse in the previous attempt (PGP for Evobrutinib and BBBP for TNG248). For good measure, another property of each molecule is then improved. Notably, at each step, the mCLM only makes minor modifications to each drug of roughly 1 building block. While the mCLM shows promising results for repairing these drugs, it is worth noting that drug discovery is a many-objective optimization problem. While we are able to generate molecules with improved toxicity relative to Evobrutinib and TNG348, as well as other properties, yet other important properties may still have been compromised. Future work may want to investigate longer reasoning chains across a wider variety of properties. For comparison, we show a (less-successful) version of this experiment using MoleculeSTM (Appendix A.3).

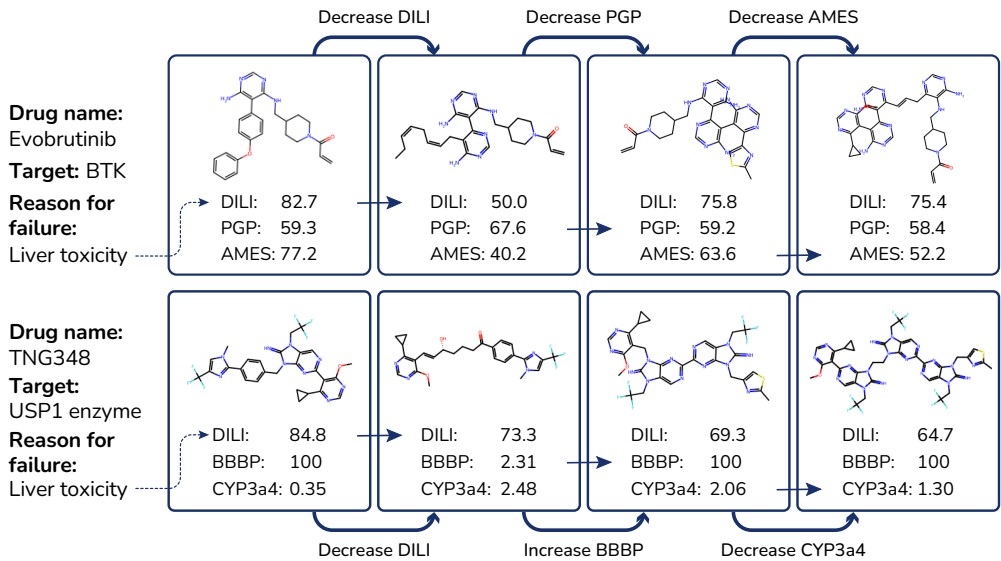

Figure 4: Examples of fallen angel property modification.

### 3.4 LIBRARY DESIGN: IDENTIFYING FUNCTIONALLY-IMPORTANT BLOCKS

As a by-product, mCLM is also able to identify blocks which are preferred for certain functions. This can help inform virtual screening campaign design (e.g., which 10 blocks should we use to get the best chemical space for BBBP?), and it can also be useful for stimulating scientific inquiry. As an example, we calculated the most frequent modifications preferred by the mCLM to improve DILI in FDA drugs (Figure 5). Notably, modification 3 resembles a newly discovered modification for reducing amphotericin B toxicity by human scientists in Maji et al. (2023).

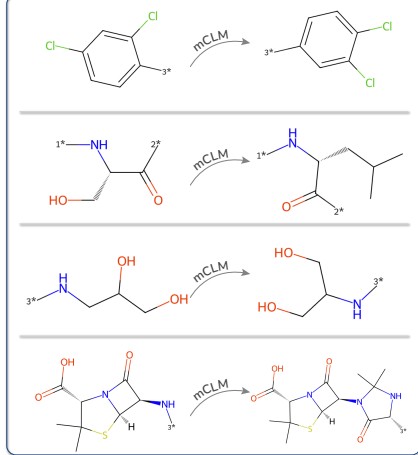

Figure 5: Four of the most frequent mCLM-proposed modifications to improve DILI in FDA-approved drugs.

## 4 RELATED WORK

Protein language models (Lin et al., 2023; Hayes et al., 2025; Wang et al., 2025b; Madani et al., 2023; Xiao et al., 2024b; Su et al., 2024; Buehler and Buehler, 2024; Wang et al., 2025a; Zhu et al., 2024; Ferruz et al., 2022) have been highly successful because proteins/peptides are tokenized at the level of amino acid building blocks (which are akin to the words in a sentence). This makes it possible to associate the sequences of these building blocks with the fold and function of proteins, and then the corresponding building blocks can be assembled into targeted sequences.

Inspired from the successes of protein LLMs, LLMs have been increasingly applied in computational chemistry (Zhang et al., 2024a;b; Fang et al., 2023; Livne et al., 2024; Pei et al., 2023; Zhao et al., 2024; 2023; Yu et al., 2024; Ye et al., 2025; Liu et al., 2023b; Li et al., 2024c; Liu et al., 2024a; Taylor et al., 2022). Recent work has further explored cross-modal modeling of molecule and language (Edwards et al., 2022b; Liu et al., 2022; Li et al., 2025b; Chang and Ye, 2024; Liu et al., 2024c). General-purpose LLMs such as GPT-4o (OpenAI, 2025a), GPT-5 (OpenAI, 2025b), Gemini-2.5-Flash (Google DeepMind, 2025), and Claude-3.5-H (Anthropic, 2025) show multimodal reasoning capabilities. However, most molecular language models still rely on atom-level (Wang et al., 2019; Ahmad et al., 2022; Zhou et al., 2023; Xia et al., 2023) or Byte-Pair Encoding on SMILES (Chang and Ye, 2024), and thus they often produce invalid tokens, misalign with chemical structures, and generate unsynthesizable molecules. To address these limitations, other approaches

incorporate functional groups and fragments into molecular representations (Li et al., 2023a; Nguyen et al., 2024c; Han et al., 2023; Jin et al., 2024; Nguyen et al., 2024a; Wang et al., 2023; Zhang et al., 2020; 2021), capturing both atomic- and group-level information. However, these methods tend to break molecules at bonds that are difficult or even impossible to form through chemical reactions. In contrast, our mCLM decomposes small molecules into function-infused and synthesis-friendly building blocks and integrates synthesis constraints early in training, akin to the way peptide/protein LLMs tokenize at the amino acid level.

Several recent papers propose chemically meaningful tokenization schemes. SAFE (Noutahi et al., 2024) modifies SMILES representations to avoid fragment splits but still inherits core issues of line notations as we described in Section 1, including validity and synthesis challenges. Group-SELFIES (Cheng et al., 2023) extends SELFIES with group-level tokens and can in principle represent our synthesis-friendly blocks. If it is used in a line notation form, however, standard LLM tokenizers will not ensure validity or synthesizability. Reasyn (Lee et al., 2025) is contemporary to our work. It focuses on synthesizable molecule projection using reinforcement learning, requiring the model to predict reactions between pairs of blocks. Unlike Reasyn, mCLM does not require the prediction of reactions, since one is guaranteed to exist between any blocks. Their task and training methodology may be relevant for future mCLM models. We also note recent efforts in learning discrete molecule representations with VQ-like mechanisms (Guo et al., 2025; Ha et al., 2025; Boget et al., 2024). However, these approaches do not guarantee synthesis compatibility and generally lack alignment with natural language. In Section H, we include comparisons with DGAE and HierVAE as additional baselines.

## 5 Conclusions and Future Work

We have developed a function- and synthesis-aware modular Chemical-Language Model (mCLM), as the first attempt to jointly model natural language sequences with modular chemical language. By design, the mCLM only generates chemical building blocks that can be iteratively assembled on robotic small molecule synthesis platforms, enabling the rapid creation of novel molecules with desired functions, all accessible by non-specialists. In the future we aim to scale mCLM to larger backbones, incorporate richer multimodal knowledge related to physical and chemical properties from 2D/3D molecular structures, protein-ligand complexes, cell lines and nucleic acid sequences to further enable the mCLM to reason on biological activity, protein docking knowledge, and individuals' genetic profiles. We plan to extend chemical reasoning to additional aspects such as filling in unspoken knowledge gaps, thinking outside of the box, System 2 thinking for counterfactual reasoning and plausibility prediction, and resolving conflicting claims. We also plan to leverage more physical constraints from simulation tools and chemical and reaction knowledge bases. In the long term, we envision the mCLM as part of a comprehensive, multi-agent, human-in-the-loop autonomous laboratory, structured around iterative cycles of reasoning, proposal, synthesis, physical testing, feedback, and reasoning to enable never-ending self-improvement and co-evolvement with human scientists.

### Reproducibility Statement

In section 3.1 and D.2.3, we describe the process and resources of developing oracle models for evaluation. The synthesis-guaranteed tokenizer is described in further detail in Section F. Sections C and D explain the data collection process and the statistics of the training dataset. Section G lists all source datasets. Model training details are described in Section E. Evaluation data from FDA-approved drugs are described in Section 3.2, which also lists the baselines we used for comparison. The full data and resources will be released upon publication.

### Acknowledgments

We would like to thank Kyunghyun Cho, Anna Hart, Hyeonjeong Ha, Gabriele Scalia, Yanru Qu, and Jeonghwan Kim for helpful discussion. This work is based upon work supported by U.S. DARPA ECOLE Program No. #HR00112390060, and NSF Molecule Maker Lab Institute, an AI Institute for Molecular Discovery, Synthesis Strategy, and Manufacturing funded by the U.S. National Science Foundation under Awards No. 2019897 and 2505932, and NSF NAIRR award. Any opinions, findings and conclusions or recommendations expressed in this material are those of the author(s) and do not necessarily reflect the views of the U.S. Government. The U.S. Government is authorized to

reproduce and distribute reprints for governmental purposes notwithstanding any copyright annotation therein. This research used the Delta and DeltaAI advanced computing and data resources, which are supported by the National Science Foundation (award OAC 2320345 and award OAC 2005572) and the State of Illinois. Delta and DeltaAI are joint efforts of the University of Illinois Urbana-Champaign and its National Center for Supercomputing Applications. We would like to acknowledge NVIDIA Corporation's contributions to this work through a grant of NVIDIA A100 Tensor Core GPUs.

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

# A    ADDITIONAL RESULTS

## A.1    SYNTHESIZABILITY

For the mCLM, all starting materials are the core building blocks, and the same collection of building blocks is used redundantly to make all the proposed structures. For this analysis, we assume that all the building blocks required for the mCLM approach are available. Given a sequence of blocks A - B - C - D in our methodology, we can assemble them with the following steps:

- Coupling reaction (A and B)

- Deprotect reaction (AB)

- Coupling reaction (AB and C)

- Deprotect reaction (ABC)

- Coupling reaction (ABC and D)

Scaling this formula to $n$ building blocks gives a requirement of $2n - 3$ reactions (synthetic steps).

On the other hand, each molecule generated by MoleculeSTM is a unique problem for synthesis, and thus cannot leverage the strengths of the block-based approach. Each generated target requires a customized synthesis solution.

However, it is important to consider an external metric for verifying the synthesis capabilities of the mCLM. In table 3, we compute validity and synthetic accessibility scores for all baselines, both overall and in terms of specific property modifications. We find that mCLM outperforms all baselines (lower SA is better).

Table 3: Synthetic Accessibility (SA) scores with RDKit validity percentages across datasets.

| Dataset | FDA | | mCLM | | MoleculeSTM | | FineMolTex | | GPT-4o | | GPT-5 | | Gemini-2.5-F | | LDMol | | Claude-3.5-H | |
|---|---|---|---|---|---|---|---|---|---|---|---|---|---|---|---|---|---|---|
| | SA | Valid (%) | SA | Valid (%) | SA | Valid (%) | SA | Valid (%) | SA | Valid (%) | SA | Valid (%) | SA | Valid (%) | SA | Valid (%) | SA | Valid (%) |
| AMES | 2.70 | 100.0 | 2.39 | 100.0 | 2.64 | 94.26 | 2.52 | 94.26 | 3.06 | 90.98 | 2.98 | 72.13 | 2.92 | 89.34 | 2.85 | 90.16 | 3.12 | 93.44 |
| BBBP | 2.70 | 100.0 | 2.44 | 100.0 | 2.67 | 93.44 | 2.61 | 95.90 | 2.67 | 88.52 | 2.87 | 78.69 | 2.69 | 90.16 | 2.89 | 94.26 | 2.90 | 93.44 |
| CYP3A4 | 2.70 | 100.0 | 2.40 | 100.0 | 2.60 | 95.90 | 2.63 | 94.26 | 2.70 | 89.34 | 2.83 | 85.25 | 2.75 | 93.44 | 2.74 | 93.44 | 2.69 | 88.52 |
| DILI | 2.70 | 100.0 | 2.38 | 100.0 | 2.62 | 94.26 | 2.59 | 93.44 | 2.73 | 93.44 | 2.75 | 81.15 | 2.70 | 90.16 | 2.80 | 87.70 | 2.79 | 88.52 |
| HIA | 2.70 | 100.0 | 2.42 | 100.0 | 2.73 | 91.80 | 2.59 | 93.44 | 2.68 | 87.70 | 2.82 | 73.77 | 2.77 | 94.26 | 2.80 | 90.16 | 2.86 | 91.80 |
| PGP | 2.70 | 100.0 | 2.45 | 100.0 | 2.63 | 93.44 | 2.59 | 94.26 | 2.81 | 90.16 | 2.85 | 86.07 | 2.79 | 82.79 | 2.87 | 93.44 | 2.75 | 90.16 |
| Mean | 2.70 | 100.0 | 2.41 | 100.0 | 2.64 | 93.85 | 2.59 | 94.26 | 2.78 | 90.02 | 2.85 | 79.51 | 2.77 | 90.03 | 2.83 | 91.53 | 2.85 | 90.98 |

Next, we select the best 3 models and consider a computationally expensive but rigorous approach. We leverage Allchemy, which is the state-of-the-art retrosynthesis software, to quantitatively measure this. We randomly sampled a representative set of 200 molecules generated by each baseline (and the original FDA molecules). We gave each molecule 30 minutes on a supercomputing cluster (which is quite exhaustive) and didn't apply a price limit for substrates (so this is really quite generous for starting materials). Scores, as reported in Table 2, show much stronger overall performance of the mCLM. Here, validity is the percent of syntactically correct molecules as measured by RDKit, synthesizability is the percent of valid molecules where Allchemy could find a synthetic path, and Makeability is the product of those two metrics. Makeability represents the percent of generated molecules which could be made in the lab.

## A.2    EXPANDED TESTING WITHOUT SYNTHESIS GUARANTEES

As a large-scale test, we apply the mCLM to improve all FDA-approved drugs consisting of at least 3 blocks (without synthesis guarantees). Note: this is just a confirmatory test, since the mCLM is intended to be used on the distribution of synthesis-guaranteed molecules (as shown in the main paper). This amounts to 430 molecular structures and 796 unseen blocks. Since most of these molecules (426/430) contain blocks that were not present in the 1,000 blocks used for training, this presents an opportunity to find how the mCLM performs out-of-distribution. Results in Table 4 show that improvement is achieved for 5/6 properties, even though the model has not seen almost half of the blocks in its vocabulary during training. Specifically, the strictest synthesis-guaranteed tokenizer covers 26.7% of compounds in our corpus. For the remainder, a rule-based tokenizer is used. Despite this fallback, mCLM demonstrates strong generalization, as it successfully incorporated GNN-derived features for the unseen modules.

Table 4: Average pharmacokinetic and toxicity properties of FDA drugs with 3 or more blocks and their proposed modifications. Note, these molecules do not have synthesis-guarantees. (↓: lower is better, ↑: higher is better.

| Property | AMES Mut. (↓) | BBBP (↑) | CYP3A4 Inhib. (↓) | DILI (↓) | HIA (↑) | PGP (↓) | Mean Improv. |
|---|---|---|---|---|---|---|---|
| **FDA Drug** | 59.5 | 37.6 | 2.0 | 66.2 | 93.2 | 66.0 | 0.00 % |
| **mCLM** | **54.0** | **41.4** | **1.2** | **55.5** | **97.6** | 68.0 | 12.87 % |

## A.3 APPLYING MOLECULESTM TO THE "FALLEN ANGELS"

As a baseline comparison, we repeat the fallen angels reasoning experiment using MoleculeSTM (Figure 6). Even though the steps 1 and 2 of MoleculeSTM's modification of TNG348 showed comparable property changes to mCLM, MoleculeSTM encounters molecule validity problems: it generates a syntactically incorrect molecule on step 1 of Evobrutinib and on step 3 of TNG348.

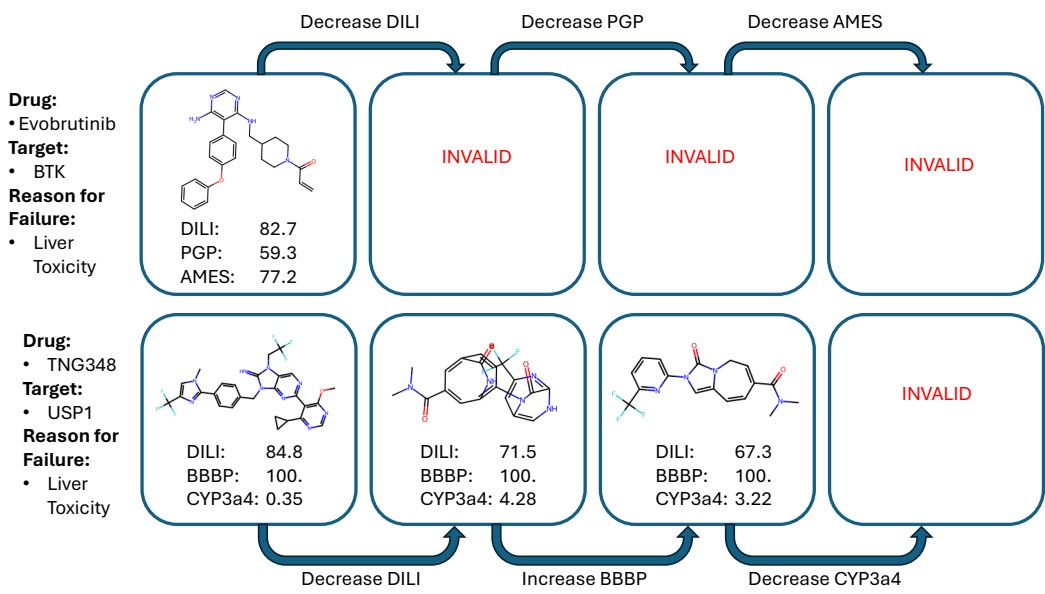

Figure 6: Examples of fallen angel property modification using MoleculeSTM.

## B CRITICAL CHEMICAL REASONING ALGORITHM

**Algorithm 1** Critical Chemical Reasoning for Molecule Design in mCLM

1: **Input:** Initial molecule $M_0$
2: **Output:** Refined molecule $M^*$
3: Initialize $M \leftarrow M_0$
4: **while** True **do**
5:     Enumerate over functions to improve in $M$ (e.g., metabolism, drug-induced organ injury, blood-brain barrier penetration)
6:     **if** No clear objective remains for improvement **then**
7:         **return** $M$
8:     **else** Select property which requires most improvement.
9:     **end if**
10:     mCLM generates a candidate modification $M'$ by replacing, adding, or removing building blocks in $M$
11:     Evaluate $M'$ with respect to desired functions
12: **end while**

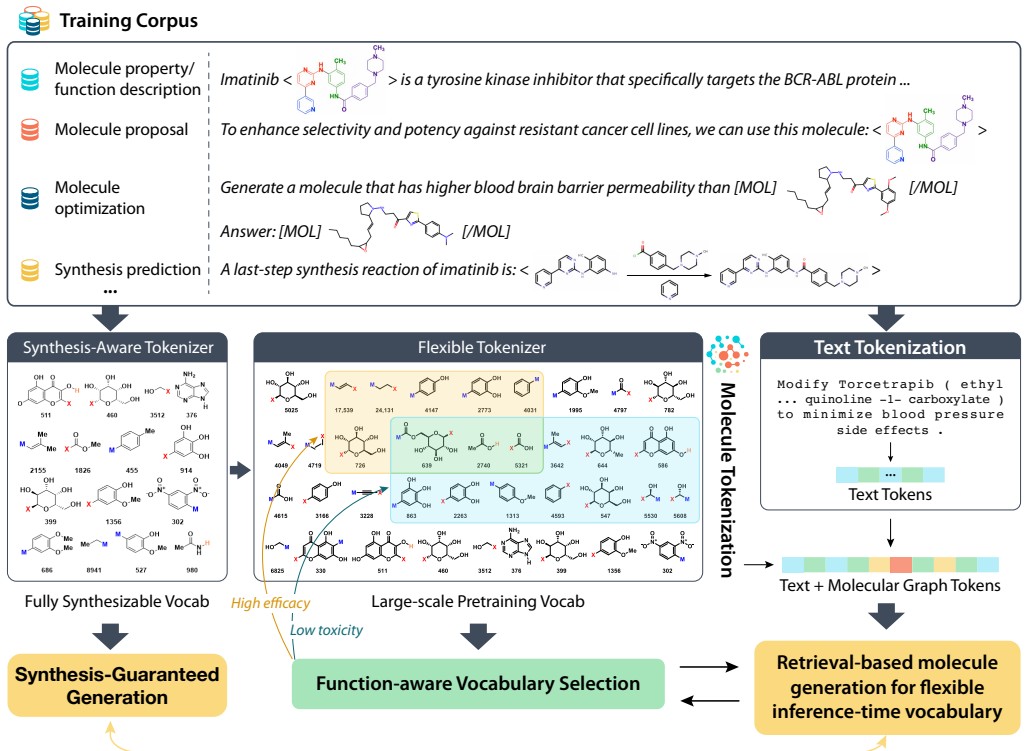

Figure 7: Overview of Data Creation.

## C  PRETRAINING DATA MIXTURE

To train the model, our goal is to cover a wide variety of different molecular functions and classes of molecules. To do so, we consider a wide variety of data sources. We use three existing instruction datasets: SMolInstruct (Yu et al., 2024), Tulu-3 (Lambert et al., 2024), and the Biomolecular text portion of Mol-Instructions (Fang et al., 2023). SMolInstruct is used to cover the standard set of chemistry LLM tasks. We supplement this with the non-overlapping portion of Mol-Instructions. Finally, we include Tulu-3 to preserve general reasoning capability.

### C.1  PROPERTY-BASED SYNTHETIC INSTRUCTION DATA GENERATION FOR CONTRASTIVE LEARNING

Additionally, we augment this data mixture with a significant amount of synthetic data created using real-world property datasets. Given our datasets, we create additional question datapoints for both regression and classification tasks, such as "Is this molecule <SMILES> ... </SMILES> blood brain barrier permeable? Yes". We also consider the opposite direction: property to molecule and multiple properties to molecule, and unconditional molecule generation. The templates used for constructing this data were created using GPT-4o; 50 were originally generated and then bad templates were removed by hand. Further, we augmented each data point with a description of the property, also written by GPT-4o.

In addition to these tasks, we also consider the molecule optimization task as described in DrugAssist (Ye et al., 2025) (e.g., given molecule A, improve property X). However, their approach has fundamental limitations due to the reliance on oracle models. These models may be out-of-distribution for molecules within our dataset, and low-quality models may propagate errors into our training data. To address this issue, we consider "activity cliffs" (Stumpfe et al., 2019; Zhang et al., 2023) within existing property datasets to train the model for molecule optimization. Activity cliffs are two molecules which are structurally similar but have large differences in a given property. We opt to base our definition of activity cliff on Murcko scaffolds (Bemis and Murcko, 1996). This methodology, which is widely used in medicinal chemistry, extracts the core subgraph, or scaffold, of a molecule. We found that this approach was much more computationally feasible on large-scale molecule datasets than approaches like fingerprint similarity (Bajusz et al., 2015).

Given nominal data (e.g., toxic, kinase inhibitor, banana smell), we look for a pair of molecules that have the same scaffold but only one molecule has the desired property. Given numerical data (e.g., solubility, power conversion efficiency, HOMO-LUMO gap), we instead look for molecules which have a property difference of 1 standard deviation. This forms the positive-negative same-scaffold portion of our data. We also consider pairs of molecules that have different scaffolds but the same property. Here, our goal is to teach the model to consider function over structure. As an example, we may ask "Propose a molecule with a different structure than [MOL] ... [/MOL] that still demonstrates anticoagulant properties. [MOL] ... [/MOL]". For a select number of property types, we use oracle models instead of ground truth data. Please see Appendix D.2.3 for more information.

## C.2 Data Filtering

Due to the large size of our dataset, we filter our data to only keep the most frequent 100k (50k end blocks and 50k mid blocks), 10k (5k and 5k), and 1000 blocks (500 and 500) for experiments. This is because there are far fewer molecule tokens compared to text (each training sample only has up to about 10). Due to Zipf's law (Zipf, 1936), in the full dataset, a single molecule token may only appear in one example in the entire training corpus. Without considerably scaling the compute budget, understanding this additional block is difficult. Overall, this filtering enables us to learn more efficiently and requires less memory resources for training the model, and allows us to better test the model architecture. Please see the section on training details for more information in Appendix E and see Appendix G for source property datasets. Table 5 gives a breakdown data quantity by instruction type for each subset of the data. Table 6 gives a breakdown of the molecule data for each subset of the data. Note that not all molecules in the "Full Data" can be represented with blocks, so we have a subset for "Only Blocks" molecules, and then molecules consisting of the most frequent "Top 100k" blocks, "Top 10k" blocks, and "Top 1000" blocks. Scaffolds are the number of unique Murcko scaffolds (Bemis and Murcko, 1996). "Caps" are end blocks, while "Mid" are middle blocks. "Synthesis-aware" is the number of synthesis-guaranteed blocks out of "Total". Data splits are subdivided into their original source: our constructed instruction data or molecule instructions from SMolInstruct.

Table 5: Dataset categories and their respective sample counts.

| Data Type | Full Data | Only Blocks | Top 100k | Top 10k | Top 1000 |
|---|---|---|---|---|---|
| **All** | 36,641,170 | 20,910,431 | 14,823,280 | 6,195,903 | 1,054,124 |
| **Existing Data Sources** | 3,117,841 | 1,208,365 | 1,049,473 | 537,674 | 149,994 |
| SMolInstruct | 2,124,738 | 215,262 | 56,370 | 14,242 | 2,300 |
| Tulu-3 | 939,343 | 939,343 | 939,343 | 469,672 | 93,934 |
| Mol-Instructions Biomol. Text | 53,760 | 53,760 | 53,760 | 53,760 | 53,760 |
| **Synthetic Real Data (Ours)** | 21,290,353 | 9,294,434 | 7,711,548 | 3,960,633 | 705,210 |
| Classification | 5,612,215 | 3,771,038 | 3,192,583 | 1,337,619 | 223,341 |
| Molecule Generation | 535,317 | 316,630 | 245,034 | 150,001 | 27,393 |
| Positive Negative Same Scaffold | 3,795,820 | 113,390 | 65,566 | 16,286 | 2,007 |
| Positive Positive Different Scaffold | 660,925 | 477,976 | 256,435 | 191,720 | 33,579 |
| Property to Molecule | 6,676,957 | 3,125,097 | 2,878,012 | 1,639,322 | 307,780 |
| Multi-Property to Molecule | 572,505 | 556,107 | 533,647 | 423,089 | 78,793 |
| Scaffold+Property to Molecule | 705,833 | 405,497 | 226,696 | 80,448 | 13,078 |
| Regression | 2,730,781 | 528,699 | 313,575 | 122,148 | 19,239 |
| **Synthetic Oracle Data (Ours)** | 12,232,976 | 10,407,632 | 6,062,259 | 1,697,596 | 198,920 |
| Classification | 2,077,327 | 1,778,874 | 1,058,754 | 299,444 | 35,168 |
| Positive Negative Same Scaffold | 72,763 | 58,645 | 25,035 | 4,935 | 469 |
| Scaffold+Property to Molecule | 1,169,407 | 925,055 | 453,087 | 113,049 | 13,642 |
| Property to Molecule | 13,257 | 13,254 | 8,841 | 906 | 2 |
| Regression | 8,900,222 | 7,631,804 | 4,516,542 | 1,279,262 | 149,639 |

# D Synthetic Data Generation

## D.1 Task Formulation for Molecular Design

To train our modular chemical language model (mCLM), we generate synthetic data that reflect realistic scenarios encountered in molecular design. These include tasks relevant to drug discovery and organic photovoltaic (OPV) materials design. Our goal is to equip the model with the ability to understand and reason over molecular representations and natural language prompts across a diverse set of tasks.

***Drug Discovery Tasks:*** In the context of drug discovery, chemists often engage in iterative and multi-objective optimization processes, where molecules are evaluated, modified, or generated based on various physicochemical

Table 6: Breakdown of molecule data by dataset category.

| Subset | Source | Molecules | | | | Molecule Tokens | | | |
|---|---|---|---|---|---|---|---|---|---|
| | | Total | Tokenized | Untokenized | Scaffolds | Cap | Mid | Total | Synthesis-Aware |
| Full Data | SMolInstruct | 1,951,205 | 1,566,030 | 385,175 | 510,464 | | | | |
| | Our data | 6,160,565 | 5,270,982 | 864,212 | 1,109,562 | | | | |
| | Total | 7,994,305 | 6,787,879 | 1,178,957 | 1,537,424 | | | | |
| Only Blocks | SMolInstruct | 459,910 | 459,910 | 0 | 145,614 | 157,204 | 59,681 | 216,885 | 50,788 |
| | Ours | 3,830,543 | 3,830,543 | 0 | 705,641 | 531,472 | 175,903 | 707,375 | 189,770 |
| | Total | 4,220,604 | 4,220,604 | 0 | 799,267 | 598,470 | 203,810 | 802,280 | 214,431 |
| Top 100k | SMolInstruct | 146,079 | 146,079 | 0 | 49,743 | 22,871 | 14,809 | 37,680 | 10,930 |
| | Ours | 2,345,519 | 2,345,519 | 0 | 429,960 | 49,941 | 49,780 | 99,721 | 27,823 |
| | Total | 2,458,034 | 2,458,034 | 0 | 457,032 | 50,000 | 50,000 | 100,000 | 28,220 |
| Top 10k | SMolInstruct | 37,686 | 37,686 | 0 | 12,373 | 4,130 | 2,534 | 6,664 | 2,618 |
| | Ours | 821,238 | 821,238 | 0 | 147,524 | 5,000 | 4,998 | 9,998 | 3,468 |
| | Total | 848,674 | 848,674 | 0 | 153,144 | 5,000 | 5,000 | 10,000 | 3,487 |
| Top 1000 | SMolInstruct | 4,867 | 4,867 | 0 | 1391 | 477 | 359 | 836 | 420 |
| | Ours | 109,554 | 109,554 | 0 | 19,639 | 500 | 500 | 1,000 | 432 |
| | Total | 112,657 | 112,657 | 0 | 20,101 | 500 | 500 | 1,000 | 432 |

and pharmacological properties. These tasks typically involve querying for specific properties, modifying structures to meet certain design criteria, or generating novel candidates that satisfy given constraints. We consider the following tasks that a chemist may perform:

- **Text-to-Molecule Generation:** Generate a molecule from a textual description of its structure or properties.

- **Property Prediction:** Given a molecule and a property of interest, predict whether the molecule possesses the property (binary classification) or the quantitative value of the property (regression).

- **Molecular Optimization:** Modify a given molecule to satisfy or improve a specific property.

- **Scaffold-Constrained Generation:** Given a scaffold and a target property, generate a molecule that satisfies both the structural constraint and the property constraint.

***OPV Material Design Tasks:*** To support broader applications of mCLM beyond drug discovery, we also incorporate tasks relevant to organic photovoltaic (OPV) material design. OPVs are an emerging class of lightweight, flexible materials used for solar energy harvesting, where the power conversion efficiency (PCE) is the primary performance metric. A typical OPV device is composed of a donor molecule and an acceptor molecule. The donor is responsible for absorbing sunlight and generating excitons (electron-hole pairs), while the acceptor facilitates charge separation and electron transport. The chemical compatibility and electronic alignment between the donor and acceptor molecules critically influence the resulting PCE. To enable learning in this domain, we define the following OPV-specific tasks

- **PCE Prediction**: Predict the PCE of a given donor–acceptor pair.

- **Donor/Acceptor Completion**: Given a donor (or acceptor) and a target PCE, generate the complementary component (acceptor or donor) that achieves the desired performance.

- **Constrained Completion with Scaffold**: Generate donor or acceptor molecules that match a given scaffold and achieve a target PCE.

## D.2 Instruction Tuning Data Generation

### D.2.1 Prompt-Answer Templates

To generate instructional data, we construct a pool of question and answer templates for each task. These templates include multiple paraphrased variants to introduce linguistic diversity and improve the model's generalization capability. During data generation, a question template is randomly sampled from the question set and populated with sample-specific content. Likewise, a corresponding answer template is sampled from the answer set to form a complete prompt–response pair. For example:

- **Question Template:** *"Given [a molecule] and [a property of interest], modify the molecule to achieve [desired property value]."*

- **Answer Template:** *"The [property] of [molecule] is [value]."*

This templating strategy allows us to produce a large number of diverse, semantically equivalent training instances that support instruction tuning across multiple molecular design tasks.

### D.2.2 Label Sources for Instruction Tuning Data

To enable diverse and meaningful pretraining for mCLM, we incorporate both experimentally derived and model-generated labels, covering a broad spectrum of molecular properties critical to chemistry, pharmacology, and materials science. This dual-labeling strategy allows the model to learn from abundant low-level molecular descriptors while also reasoning over high-level functional and biological endpoints.

***Low-Level Molecular Properties from ChEMBL:*** For foundational chemical descriptors, we leverage the ChEMBL25 database—a comprehensive bioactivity resource containing approximately 2 million compounds with rich structural and physicochemical annotations. ChEMBL25 serves as an abundant and reliable source of labels for low-level properties that are widely used in cheminformatics pipelines. From this corpus, we select a core set of descriptors that are most informative for molecular design: Hydrogen bond acceptors (HBA), Hydrogen bond donors (HBD), LogP (octanol–water partition coefficient), Molecular weight (MolWt), Number of aromatic rings, Number of rotatable bonds, Topological polar surface area (TPSA). These descriptors are inexpensive to compute and provide critical insights into molecular solubility, permeability, and synthetic feasibility—making them essential for early-stage screening and property-based filtering.

***High-Level Molecular Properties via Oracle Labeling:*** In addition to low-level properties, we aim to expose the model to high-level functional endpoints that capture complex biological phenomena. Such endpoints are central to pharmacokinetics, drug safety, and efficacy, but they are rarely available in large quantities due to the high cost of experimental validation. Consequently, labeled datasets for these tasks are limited in size and diversity. To address this challenge, we employ the oracle ensemble models to generate synthetic labels for a curated set of ADMET tasks.

### D.2.3 Ensemble Oracle Model:

To generate high-quality synthetic labels for downstream tasks, we construct oracle models focused on ADMET property prediction. The oracle models we use for ADMET assessment are state-of-the-art, high-performing models trained on experimental benchmark datasets with strong reported correlation to assay measurements, as shown in Table 7 below. This setup is highly meaningful in computational drug discovery (especially compared to rougher and less meaningful QED scores) and is designed to emulate digital screening pipelines used in industry.

We select tasks from the Therapeutics Data Commons (TDC) benchmark [†] using the following criteria:

- **Relevance to Drug Discovery:** The task must reflect a critical aspect of drug efficacy or toxicity.
- **Predictability:** The task must be reliably predictable using existing models. Specifically, we evaluate all 22 ADMET-related classification tasks in TDC and retain only those where standard models achieve an area under the ROC curve (AUC) greater than 0.80. This ensures the synthetic labels are sufficiently accurate for training purposes.

Based on these criteria, we select six tasks:

- **AMES** (mutagenicity),
- **BBBP** (blood-brain barrier permeability),
- **CYP3A4** inhibition (metabolism),
- **DILI** (drug-induced liver injury),
- **HIA** (human intestinal absorption),
- **PGP** (P-glycoprotein substrate classification).

TDC provides predefined scaffold-based data splits with an 8:1:1 ratio for train, validation, and test sets. This splitting strategy ensures that structurally dissimilar compounds are separated across subsets, encouraging generalization to novel scaffolds.

Although TDC provides leaderboards for these tasks, many top-performing entries lack reproducible code or working implementations. For instance, the authors of one of the top submissions explicitly acknowledge on GitHub that their code is not runnable.[‡] Therefore, we opt to use robust foundation models—FARM, ChemBERTA-2, and a GNN—for ensemble learning.

- **FARM** (Nguyen et al., 2024a): A SMILES-based BERT model trained with functional group-aware tokenization.

---

[†] https://tdcommons.ai/benchmark/admet_group/overview/
[‡] https://github.com/maplightrx/MapLight-TDC

- **ChemBERTA-2** (Ahmad et al., 2022): A large-scale transformer model trained on millions of canonical SMILES sequences.
- **GNN** (Edwards et al., 2024a): A graph neural network trained on molecular graphs with atom- and bond-level features.

To build the ensemble, we use each model as a feature extractor. The extracted features are concatenated and passed through a fully connected layer for final prediction. This ensemble approach is stacking, where multiple base learners feed into a meta-learner. For each task, we select a threshold that maximizes the F1 score on the validation set. This threshold is then used to binarize the predicted logits into class labels. The performance of our ensemble model across the selected tasks is summarized in Table 7.

Table 7: Performance (AUC) of individual models and the ensemble across six selected ADMET tasks.

| Model | AMES | Pgp | DILI | BBBP | CYP3A4 | HIA |
|---|---|---|---|---|---|---|
| FARM (Nguyen et al., 2024a) | 0.88 | 0.89 | 0.79 | **0.94** | 0.88 | 0.92 |
| GNN (Edwards et al., 2024a) | 0.75 | 0.78 | **0.86** | 0.79 | 0.80 | 0.81 |
| ChemBERTa-2 (Ahmad et al., 2022) | 0.86 | 0.89 | 0.81 | 0.93 | 0.86 | **0.99** |
| **Ensemble** | **0.89** | **0.91** | 0.84 | 0.93 | **0.89** | **0.99** |

# E  TRAINING PROCEDURE

We employed Qwen2.5-3B (Yang et al., 2024) as the starting LLM for building the mCLM. Generally, we followed the training procedure from LLaVa (Liu et al., 2023a; 2024b; Li et al., 2024a). We used a two-layer MLP with PReLU activation (He et al., 2015) as an adapter into the LLM input/output from the GNN. We selected an initial learning rate of 1e-5 for the full model and 1e-6 for the adaptor and LM heads. Further, we used a cosine annealing schedule with a minimum of 1e-6 and 2000 linear warmup steps; AdamW (Loshchilov and Hutter, 2017) optimizer was employed. The model was trained on 4 A100 80GB GPUs in bfloat16 precision.

We found that the model learned the molecule tokens much slower than the text (there is usually a 10x difference in loss value). Molecule tokens are rarer and show up less in the training data. Because of this, we decided to separate the molecule classifier head and the language classifier head. We used a standard autoregressive language modeling loss for both, and we averaged these two losses for the final loss value. The main part of our training experiments focused on minimizing the molecule loss, since the text loss was easy to optimize. Further, we found PEFT (Hu et al., 2021) was not sufficient to adapt to molecules, so full finetuning was required. Roughly 10-50 examples from each synthetic (data source, task) pair were put into a validation set.

To initialize the GNN weights, we employed the MolCLR (Wang et al., 2022b) unsupervised contrastive learning technique. We used AugliChem (Magar et al., 2022) for the augmentations: random atom masking, random bond deletion, and motif removal. One of these augmentation was selected uniformly at random for each data point. The GNN was initialized using a batch size of 128 and lr of 1e-4 with a cosine schedule. The model was trained on all 800k blocks in the full data until convergence on a validation set. We tested embedding dimensions between 16 and 4,096 and found 128 dimensions to be sufficient while minimizing total memory cost. This was necessary because we stored the entire embedding matrix in GPU memory, which was much faster, but consumed about 20GB VRAM. Doing so allowed us to train without the GNN during our pretraining process, which is considerably more efficient. We note the GNN can then be finetuned along with the rest of the mCLM during finetuning to new types of molecules or specific tasks. While we did consider a sampled softmax to train the mCLM, we found this to limit the learning of the model.

For training the mCLM, we used two stages for pretraining. First, we trained for 1 epoch with everything frozen except the adaptor, to allow the adaptor to adjust to the LLM representation space. For the second stage, we trained for 5 epochs with only the GNN embeddings frozen. As discussed in the training data mixture section C, we used the most frequent 1000 building blocks as our vocabulary.

After pretraining, we finetune the mCLM to standardize it's outputs for our experiments. During pretraining, we train for robustness by using a wide variety of responses (e.g., for BBBP prediction we might respond "It is restricted from entering the central nervous system" instead of 'No'). For finetuning, we train with standardized responses for our desired tasks (e.g., "Generate a molecule that has higher blood brain barrier permeability than [MOL] ... [/MOL].", "[MOL] ... [/MOL]". Due to our downstream tasks, we finetuned exclusively on the molecule optimization task for 5 epochs over 100k examples for each property. We trained using the same procedure as the pretraining stage, but we selected the best model using validation loss.

# F  TOKENIZER DETAILS

## F.1  SYNTHESIS-GUARANTEED TOKENIZER DETAILS

The synthesis-guaranteed tokenizer disconnects the molecule only at bonds that can be formed by a pre-determined small set of reactions, preferably only those that can be performed in an automated manner. For instance, if amide bond formation is defined as available, the tokenizer will be able to disconnect amide bonds in the molecule of interest. Up to this point, the protocol is synonymous with classical computational retrosynthesis, but there is a fundamental difference. Namely, the sets of reactions suitable for automated synthesis is very limited – in fact, state-of-the-art synthesis machines utilize only three types of disconnections (amide bond formation, as well as Suzuki and Buchwald-Hartwig couplings). This places very stringent requirements on the groups that can be present in the disconnected blocks – for instance, when the disconnection (say, Buchwald-Hartwig coupling) yields an amine functionality on one of the blocks, this block cannot contain any groups that during the anticipated uses of this block would present a synthetic incompatibility. In the most trivial case, the block cannot contain another unprotected amine because after the Buchwald-Hartwig disconnection, the block would feature two amines which, in turn, would present competing reactive sites (in the Buchwald-Hartwig synthesis but also in the formation of amide bonds). Therefore, the tokenizer performs retrosynthetic operations while simultaneously checking if they do not lead to blocks with functional groups presenting competing reactivities. Only disconnections avoiding such problems are allowed. This then guarantees that when the corresponding blocks are used to make other molecules, they give only the selective synthesis outcomes. The tokenization process maintains the integrity of the molecules: each molecule can be reconstructed exactly from the building-block sequence, whether produced by tokenization or generated by the mCLM. Each tokenized building block contains placeholder atoms (e.g., [1*], [2*], [3*]) that encode connection points. These placeholders allow us to preserve the full structural integrity of the original molecule, enabling lossless reconstruction from a sequence of tokens. For example, as shown in Figure 1, the drug imatinib with SMILES string

```
Cc1ccc(cc1Nc2nccc(n2)c3cccnc3)NC(=O)c4ccc(cc4)CN5CCN(CC5)C
```

is tokenized into:

```
[ "[3*]c1cccnc1",
  "[2*]c1ccnc(N[1*])n1",
  "[1*]Nc1ccc(C)c([2*])c1",
  "[3*]C(=O)c1ccc(CN2CCN(C)CC2)cc1" ]
```

Here, [3*] connects to [2*], [1*] connects to [2*], and so on. This process is fully deterministic and reversible. The placeholder atoms and their positions are also encoded by the GNN encoder, allowing the model to reason about both structural and functional context during generation.

Following the connection rules embedded in the placeholders, the original molecule can be fully reconstructed with the following steps:

```
1. ["[3*]c1cccnc1", "[2*]c1ccnc(N[1*])n1"] → "[1*]Nc1nccc(-c2cccnc2)n1"
   (following the rule [3*] connects to [2*].
2. ["[1*]Nc1nccc(-c2cccnc2)n1", "[1*]Nc1ccc(C)c([2*])c1"]
    → "[1*]Nc1ccc(C)c(Nc2nccc(-c3cccnc3)n2)c1"
   (following the rule [1*] connects to [2*]).
3. ["[1*]Nc1ccc(C)c(Nc2nccc(-c3cccnc3)n2)c1", "[3*]C(=O)c1ccc(CN2CCN(C)CC2)cc1"]
    → "Cc1ccc(NC(=O)c2ccc(CN3CCN(C)CC3)cc2)cc1Nc1nccc(-c2cccnc2)n1"
   (following the rule [1*] connects to [3*]).
```

## F.2  MOLECULE TOKEN EXAMPLES

As discussed in E, we pretrained our GNN encoder using MolCLR (Wang et al., 2022b) on all 800k blocks in our full pretraining dataset. Figures 8-13 shows six examples of random blocks and their 5 nearest neighbors in our dataset. The similarity of the blocks and their neighbors shows the high quality representations which were obtained.

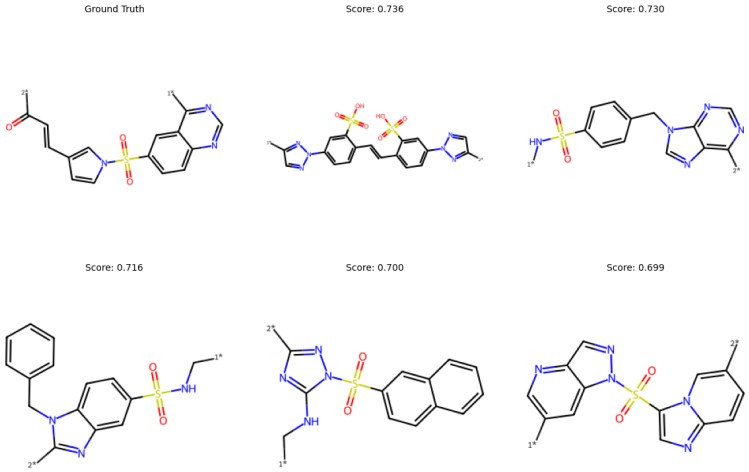

Figure 8: Example token from pretraining dataset. Similarity above each molecule is ECFP4 Fingerprint Tanimoto similarity from the ground truth molecule.

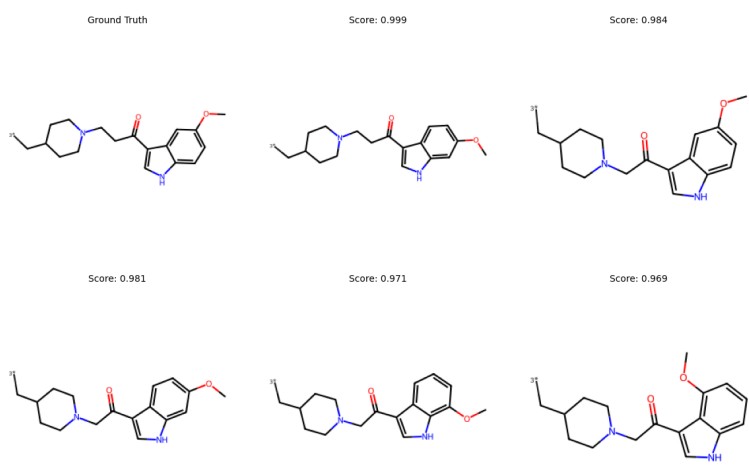

Figure 9: Example token from pretraining dataset. Similarity above each molecule is ECFP4 Fingerprint Tanimoto similarity from the ground truth molecule.

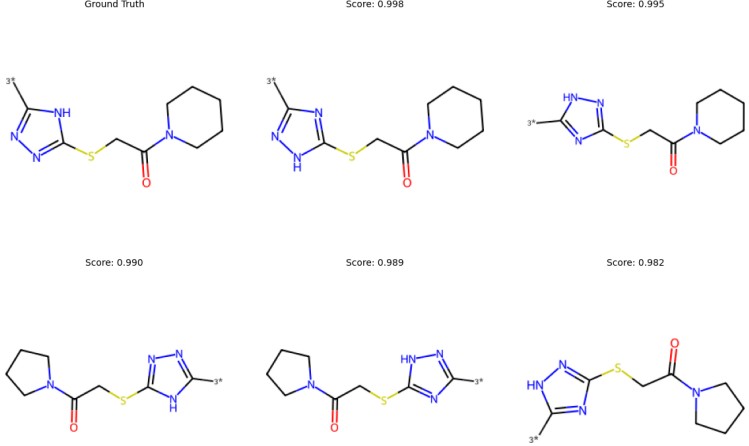

Figure 10: Example token from pretraining dataset. Similarity above each molecule is ECFP4 Fingerprint Tanimoto similarity from the ground truth molecule.

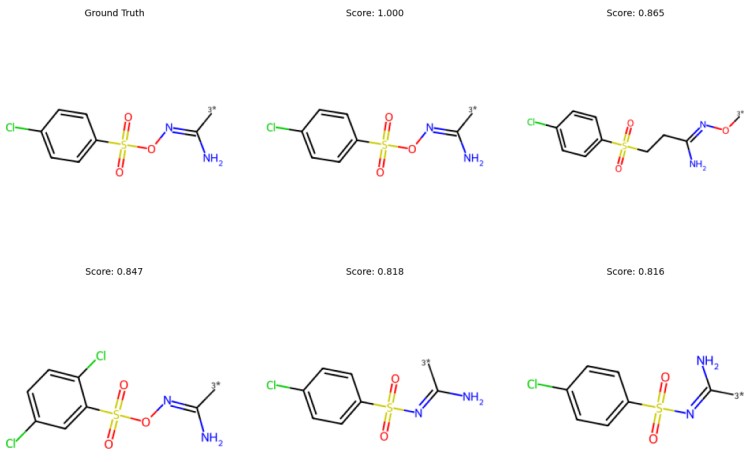

Figure 11: Example token from pretraining dataset. Similarity above each molecule is ECFP4 Fingerprint Tanimoto similarity from the ground truth molecule.

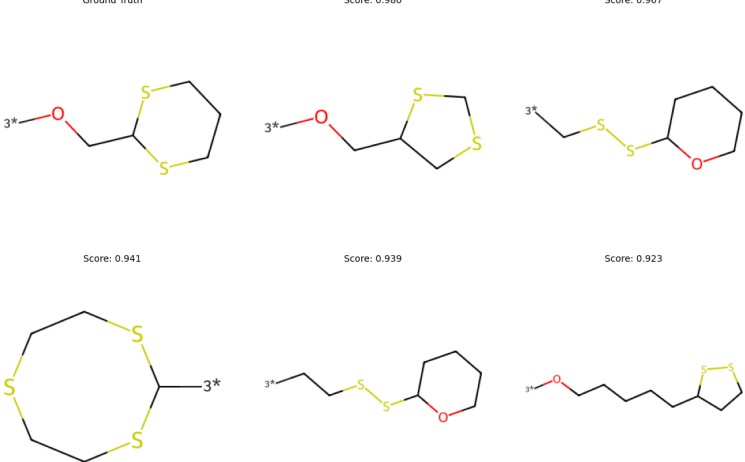

Figure 12: Example token from pretraining dataset. Similarity above each molecule is ECFP4 Fingerprint Tanimoto similarity from the ground truth molecule.

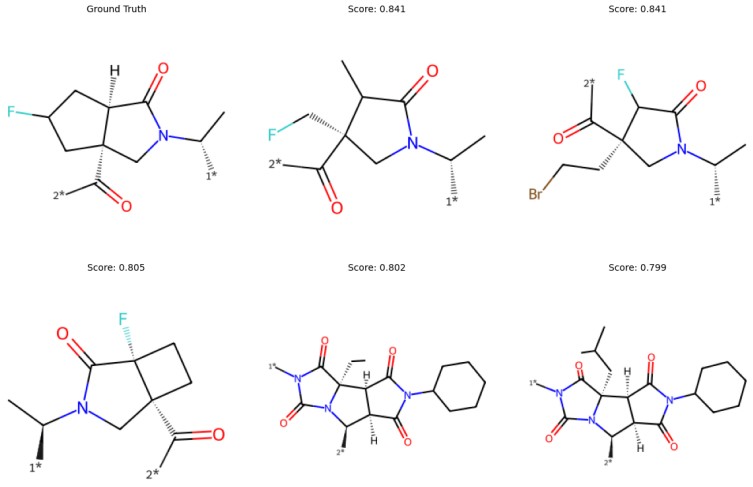

Figure 13: Example token from pretraining dataset. Similarity above each molecule is ECFP4 Fingerprint Tanimoto similarity from the ground truth molecule.

## G    Source Datasets and Databases

- Leffingwell odors (Sanchez-Lengeling et al., 2019)
- BACE (Wu et al., 2018)
- Flashpoint (Sun et al., 2020)
- MUV (Wu et al., 2018)
- Tox21 (Wu et al., 2018)
- AMES (Huang et al., 2021)
- Bioavailability (Huang et al., 2021)
- Caco2 (Huang et al., 2021)
- Carcinogens (Huang et al., 2021)
- cav3_t (Huang et al., 2021)
- choline_transporter (Huang et al., 2021)
- clearance hepatocyte (Huang et al., 2021)
- CYP1A2 (Huang et al., 2021)
- CYP2C9 (Huang et al., 2021)
- CYP2D6 (Huang et al., 2021)
- CYP3A4 (Huang et al., 2021)
- DILI (drug induced liver toxicity) (Huang et al., 2021)
- Half_life (Huang et al., 2021)
- hERG (Huang et al., 2021)
- HIA (human intestinal absorption) (Huang et al., 2021)
- Hydration free energy (Huang et al., 2021)
- kcnq2 (Huang et al., 2021)
- LD50 (Huang et al., 2021)
- Lipophilicity (Wu et al., 2018)
- m1_muscarinic (Huang et al., 2021)
- OPV data (Nguyen et al., 2024b; Lopez et al., 2016)
- orexin_receptor (Huang et al., 2021)
- PAMPA_NCATS (Huang et al., 2021)
- Broccatelli (Broccatelli et al., 2022)
- potassium_ion_channel (Huang et al., 2021)
- PPBR (Plasma Protein Binding Rate) (Huang et al., 2021)
- Pubchem logP (Kim et al., 2023)
- SARSCoV2_3CL (Huang et al., 2021)
- SARSCoV2_vitro (Huang et al., 2021)
- serine_threonine_kinase_33 (Huang et al., 2021)
- Skin Reaction (Huang et al., 2021)
- Solubility_AqSolDB (Huang et al., 2021)
- tyrosyl-dna_phosphodiesterase (Huang et al., 2021)
- VDss (volume of distribution at steady state) (Huang et al., 2021)
- Molecule Property Cliff Datasets (30+ datasets) (Van Tilborg et al., 2022)
- Chemical Function (CheF) (Kosonocky et al., 2023)
- ChemFOnt: the chemical functional ontology resource (Wishart et al., 2023)
- Pubchem properties (Kim et al., 2023)
- FreeSolv (Wu et al., 2018)
- QM8 (Wu et al., 2018)

- QM9 (Wu et al., 2018)
- Thermosol (Wu et al., 2018)
- ESOL (Estimated SOLubility) (Wu et al., 2018)
- Lipo (Wu et al., 2018)
- BBBP (Gaulton et al., 2012)
- ClinTox (Wu et al., 2018)
- HIV (Wu et al., 2018)
- SIDER (Wu et al., 2018)
- Forward synthesis (USPTO) (Yu et al., 2024)
- Retrosynthesis (Yu et al., 2024)
- CheBI-20 (Edwards et al., 2021)
- L+M-24 (Edwards et al., 2024c)
- HBA (Gaulton et al., 2012)
- HBD (Gaulton et al., 2012)
- MolWt (Gaulton et al., 2012)
- NumAromaticRings (Gaulton et al., 2012)
- rotatable_bonds (Gaulton et al., 2012)
- TPSA (topological polar surface area) (Gaulton et al., 2012)

## H    ADDITIONAL BASELINES

We conducted additional comparisons with two relevant baselines: DGAE (Boget et al., 2024) (a discrete graph autoencoder using vector quantization) and HierVAE (Jin et al., 2020) (a hierarchical VAE for molecular graph generation). Property scores are in Table 8 and sythesizability scores in Table 9. A unique HierVAE model is trained individually for each property using data from the mCLM corpus, giving it an unfair advantage in per-property optimization, while mCLM operates via instruction prompting. Interestingly, HierVAE performs quite well on synthesizability; we believe that to some extent, this is because HierVAE is trained using molecules constructed from mCLM tokenizer outputs. Thus, it is able to take advantage of the distribution of molecules created by mCLM preprocessing, which likely increases it's synthesizability performance. Also interestingly, we find a disagreement between the less-rigorous SAscore and actual retrosynthesis metrics, where mCLM outperforms significantly on the more grounded retrosynthesis score. We speculate this is because SAScore prefers simpler molecules or those with simpler substructures, whereas retrosynthesis tools evaluate actual synthetic feasibility rather than only based on structural patterns. However, we note that the difference in synthesizability is still small. Notably, Allchemy may also simply be too strong for the molecules in this distribution, which reduces the dynamic range of the metric. These specific FDA-approved drugs may be too simple, so the dynamic range of molecules predicted synthesizable by Allchemy is quite restricted. For example, even models like MoleculeSTM have > 90% synthesizability when editing these molecules. Future work may need to compare models on more difficult distributions of molecules to increase the dynamic range.

Additionally, we find that HierVAE is effective for optimizing properties (Table 8), especially properties that already have good scores, such as CYP3A4 and HIA. Still, mCLM outperforms it on average.

Table 8: Comparison of pharmacokinetic and toxicity property scores between mCLM and HierVAE. Note that HierVAE is fine-tuned separately for each property, whereas mCLM is used in a instruction-following setting.

| Model | AMES ($\downarrow$) | BBBP ($\uparrow$) | CYP3A4 ($\downarrow$) | DILI ($\downarrow$) | HIA ($\uparrow$) | PGP ($\downarrow$) | Avg. Improv. |
|---|---|---|---|---|---|---|---|
| HierVAE | 48.2 | 66.4 | **1.2** | 53.1 | **99.7** | **64.3** | 10.5% |
| mCLM (Ours) | **44.4** | **85.2** | 1.4 | **53.7** | 98.99 | 64.4 | **15.0%** |

We would also like to explain the technical differences between mCLM and vector-quantized methods, and why we adopt our current approach instead of a vector-quantization technique. Vector-quantized autoencoders (e.g., VQ-VAE) are commonly used to build discrete vocabularies for downstream models. However, in the context of molecular design, these learned codebooks lack chemically grounded guarantees. In contrast, our vocabulary consists of synthesis-verified building blocks, derived from retrosynthesis constraints and domain knowledge, ensuring that all generated molecules are compatible with automation-friendly synthesis. Further,

Table 9: Synthetic accessibility (SA) (Ertl et al., 2009), validity, and retrosynthetic results for HierVAE. Synthesizability is the percent of valid molecules where a retrosynthetic route was found. Makeability is the overall percent of generations which can be synthesized (Makeability =Valid $\times$ Synth.).

| Model | SA ($\downarrow$) | Validity (%) | Synthesizability (%) | Makeability (%) |
|---|---|---|---|---|
| FDA | 2.70 | **100.0** | 98.11 | 98.11 |
| mCLM (Ours) | 2.43 | **100.0** | **98.23** | **98.23** |
| HierVAE | 2.35 | 99.70 | 95.83 | 95.55 |

applying vector quantization in this domain risks collapsing chemically distinct structures into the same token, reducing interpretability and limiting extension to new blocks at inference-time. Instead, we use a GNN-based encoder pretrained on chemical properties to produce semantically meaningful embeddings of molecular blocks, which are aligned with a language model via adapter layers. This design allows mCLM to retain both chemical fidelity and generation flexibility, while supporting plug-and-play extensions and instruction-based editing. In Table 10, we compare synthesizability of the DGAE and mCLM on the QM9 dataset. Here, we leverage an existing DGAE model trained on QM9, released by the authors (Boget et al., 2024). The mCLM is restricted to use synthesis-guaranteed QM9 blocks. In this experiment, the difference is quite striking– the use of our specialized blocks instead of a codebook nearly doubles the synthesizability of generated molecules.

Table 10: Synthesizability comparison using Allchemy on QM9 molecules.

| Model | Validity (%) | Synthesizability (%) | Makeability (%) | SA Score ($\downarrow$) |
|---|---|---|---|---|
| DGAE (VQ-based) | **100.0** | 62.0 | 62.0 | 4.46 |
| mCLM (Ours) | **100.0** | **100.0** | **100.0** | **2.23** |

## I  ABLATION STUDIES

As detailed in the main text, we performed two ablations of the mCLM block encoding process. First, we trained a SMILES-based version of the mCLM without a GNN (No GNN), still using a Qwen-3B backbone. Second, we conducted a test of the mCLM using blocks extracted from the BRICS algorithm instead of from our tokenization method (No Synth. Tokenizer). These showed significantly degraded property modification capabilities compared to the mCLM.

We also did an ablation study of using Llama-3.2-3B as the backbone for mCLM. We find that, as also noted by others in the community, with the same backbone size, Qwen works better when fine-tuned on downstream tasks than Llama. Specifically, we find evidence that the Llama-3.2-3B-mCLM is overfitting compared with Qwen-2.5-3B-mCLM, evidenced by a lower training loss but higher validation loss (3.702 v.s. 3.620). We also evaluated the property optimization capability of Llama-3.2-3B-mCLM as follows:

Table 11: Comparison of pharmacokinetic and toxicity property scores between using Qwen-2.5-3B and Llama-3.2-3B as backbones.

| Model | AMES ($\downarrow$) | BBBP ($\uparrow$) | CYP3A4 ($\downarrow$) | DILI ($\downarrow$) | HIA ($\uparrow$) | PGP ($\downarrow$) | Avg. Improv. |
|---|---|---|---|---|---|---|---|
| mCLM (Llama-3.2-3B) | 48.2 | 59.4 | 1.7 | 59.0 | 98.1 | **63.9** | 2.82% |
| mCLM (Qwen-2.5-3B) | **44.4** | **85.2** | **1.4** | 53.7 | **98.99** | 64.4 | **15.0%** |
| mCLM (No GNN) | 50.4 | 46.0 | 2.5 | **50.2** | 97.9 | 71.7 | -7.53% |
| mCLM (No Synth. Tokenizer) | 48.7 | 55.0 | 2.9 | 54.9 | 98.3 | 68.3 | -19.0% |

