# OpenReview forum: "mCLM: A Modular Chemical Language Model that Generates Functional and Makeable Molecules"
_ICLR.cc/2026/Conference — ICLR 2026 Oral_

### Official Review · Reviewer_XT6a · 2025-10-27

**Soundness:** 3
**Presentation:** 3
**Contribution:** 3
**Rating:** 6
**Confidence:** 5

**Summary:**

This paper proposes mCLM, a modular chemical-language model that helps the translation between natural language description and molecular blocks. Different from previous chemical LLMs using SMILES or SELFIES as their inputs, mCLM learns a new molecular language that is more effective in property prediction and better in synthesis. Experimental results show that with 3B parameters, mCLM can achieve much better synthetic accessibility and property scores than LLMs like GPT-5.

**Strengths:**

1. The idea of proposing a new molecular language and tokenizing molecules into substructures is novel and helps LLMs better focus on the structural patterns of molecules that will affect the molecular properties.
2. The performance of mCLM is significantly much better than previous baselines. With only 3B parameters, mCLM is also much more efficient than LLMs like GPT-5.
3. This paper is overall well-written and easy to follow.

**Weaknesses:**

1. The proposed molecular vocabulary lacks sufficient details regarding its implementation. Furthermore, it is worth investigating whether the decomposition of molecules into these substructures could potentially compromise their structural integrity.
2. The molecular tokenization in mCLM utilizes GNNs to generate embeddings for the building blocks. However, the study does not explore or compare this approach against more advanced or alternative methods, such as VQ-VAE, which could potentially offer superior performance.
3. The evaluation of mCLM is confined to the Qwen-3B base model. To assess the scalability and generalizability of the proposed method, it would be beneficial to test it across different models of varying sizes and architectures.
4.  The experimental section primarily benchmarks against other LLMs. While some of the chemical LLMs used as baselines may be outdated, the evaluation notably omits comparisons with established GNN baselines, which are crucial for a comprehensive assessment.
5. The proposed molecular property optimization task could be significantly strengthened by incorporating more relevant and diverse datasets into the evaluation.

**Questions:**

Please address my concerns in the Weaknesses.

---

> ### Author Response · Authors · 2025-11-21
> **Response to Reviewer XT6a**
>
> Thank you for your positive review on our novel tokenization approach. We’re glad you found our idea of “tokenizing molecules into substructures” helpful for modeling molecular properties and synthesis. We also appreciate your recognition that mCLM achieves “much better synthetic accessibility and property scores” while being more efficient than large-scale LLMs like GPT-5.
>
>
>
> **Weakness1**: Thank you for raising this point. We would be happy to clarify any specific aspects of the vocabulary construction that you find lacking. Overall, we follow the procedure in Figure 3. Appendix F provides further details on the synthesis-guaranteed tokenizer, and Table 6 contains a detailed breakdown of the vocabulary composition.
>
> Regarding your concern about structural integrity: the decomposition process is designed to maintain a 1-to-1 mapping between each molecule and its corresponding sequence of building blocks. This means that it is possible to convert from a building block sequence back to the original molecule graph without losing information. Hence, the structural integrity is fully preserved.
>
> **Weakness 2**: Using a VQ-VAE for tokenization is a pertinent suggestion, since it is a popular approach within vision-language models. However, VQ-VAEs are primarily designed for continuous input domains (e.g., vision or audio) where VQ discretizes continuous inputs (such as images) into tokens which can be used in a LLM. In our setting, on the other hand, molecular building blocks *already* form a discrete vocabulary. Applying vector quantization in this setting would risk collapsing multiple chemically distinct blocks into the same code. This would reduce the model’s ability to distinguish important structural differences and make it difficult to incorporate new blocks at inference-time. We will add this discussion into the final version of the paper.
>
> **Weakness 3**: This is a practical suggestion. We agree that evaluating mCLM across models of different sizes and architectures would provide useful insights into scalability and generalizability. Due to resource constraints, we use Qwen-3B as a practical and representative starting point. We would like to note that we conducted early experiments using Llama3.2-3B, but it had higher validation loss than Qwen. Testing larger models or additional architectures would require substantially more computational resources than affordable for an academic lab, so we leave this as future work. To address this point, we will expand the discussion in the camera-ready version to clarify this design choice and outline scaling options.
>
>
> **Weakness 4**: Comparing against GNNs is a practical suggestion since they are a staple in molecular tasks. However, GNNs alone aren't generative models, so we can't directly compare against them for molecule generation. From our baselines, we instead consider autoregressive SMILES generative models and diffusion-based models. Notably, however, GNNs are used for training the CLIP-style models used in MoleculeSTM and FineMolTex (although the GNNs aren’t used for the actual generation).
>
> **Weakness 5**: Thank you for the suggestion. In our current work, we consider FDA-approved drugs and fallen-angel compounds because these datasets have direct real-world impact and are well studied. We agree that broader property coverage is valuable. However, we argue it is critical to ensure our evaluations are rigorous and can be trusted. We initially considered 22 relevant properties from Therapeutics Data Commons, spanning absorption, distribution, metabolism, excretion, and toxicity, among others. From this large starting point, we selected six properties because the oracle model has sufficient predictive performance (detailed in Appendix D.2.3). This setup allows us to evaluate the optimization capability of the mCLM while ensuring that the results are trustworthy and clinically meaningful. Going forward, we will incorporate additional datasets as high-quality property predictors become available, and we view this as an important future work.
>
>
> Thanks again for your insightful comments– we will work to better clarify and address these in the final version of the paper to improve its accessibility to a broader audience.

---

> ### Comment · Reviewer_XT6a · 2025-11-21
>
> I would like to thank the authors for their rebuttal, but I do not feel respected that almost all my questions are skipped without substantial experiments.
>
> Regarding W1, I am not asking how the tokenizer is composed of, but about how you generate these motifs and can these motifs really work as the minimal functional units? Regarding structural integrity, I am asking how do you represent the connections among these motifs? Does this tokenization keep the information of edges?
>
> Regarding W2, I can not agree with this response. When trying use LLMs or GNNs to model molecules, there is a hypothesis that molecules can be mapped to an embedding space. And inherently, what mCLM does is to use these motifs instead of the codebook.
>
> Regarding W3, if 3B-level model is acceptable, why not try llama-3B?
>
> Regarding W4, I can provide a direct baseline, JT-VAE, for your reference.
>
> I hope the authors could take endeavors to really pay attention to my comments. Thanks!

---

> ### Author Response · Authors · 2025-11-24
> **Follow-Up Response to Reviewer XT6a (Part 1/3)**
>
> Thank you for the further clarification of your comments. We appreciate your suggestions and have revised our responses to address your concerns more concretely. In our efforts to get responses to all the reviewers in time, we had prioritized certain experiments given our compute budget. Now with the additional time we have had a chance to run all of the experiments you recommended. We have completed two of them (although there were implementation challenges in certain cases, please see details below) and in both cases, the results further support the unique capabilities of mCLM. Further, we are currently conducting an ablation on Llama. We are happy to continue the discussion and answer any further questions you may have.
>
> **About W1-related questions**
>
> *Subquestion W1-1: About the motif generation process.* We thank the reviewer in raising a question about the generation details. We collected roughly 8 million molecules from a wide variety of sources (Appendix G), and among these roughly 6.8 million were tokenizable by our full tokenization approach (Figure 3). This resulted in ~200k motifs/building blocks (Table 6). Thus, motifs are discovered from real data distributions (using specific rules and retrosynthesis constraints). We will make this explicit and clear in the revision.
>
> *Subquestion W1-2: Regarding whether motifs “serve as minimal functional units”:*
> This is a great point, and it also happens to be very property specific. In some extreme cases, the individual blocks or motifs are very important. Examples of this include 1) toxic metabolites, such as furans [12] or heterocyclic amines [13], where a single functional group will make almost any molecule toxic, 2) morpholine [14], which increases the solubility of molecules, or 3) hinge binders [15, 16], which are critical for binding with kinases. Figure 4 in our paper gives a good illustration of such single-block modifications.
>
> It is true that in many cases, the functional properties of whole molecules will be more than the sum of their parts. That said, there is abundant evidence that the parts matter, and that the rules about how they combine in sequences to yield emergent functional properties can be learned. This same concept has been clearly demonstrated with proteins, DNA, and RNA, leading to highly impactful protein language models (PLMs) (such as Alpha Fold[1] ), DNA language models (such as GROVER[2]), and RNA language models (such as RiNALMo[3]).
> With respect to the specific question about “serve as minimal functional units”, mCLM transfers the functional information associated with each drug molecule to the building blocks contained within it. When building blocks redundantly appear in multiple drugs, and those drugs have similar properties, the (latent space) functional encoding of those functions with those blocks is strengthened. In this way, mCLM encodes functional information into the building block representations, which it then leverages to generate new molecules with new predicted functions. As shown in our experiments in the paper, this approach appears to be very promising for successfully generating whole molecules predicted to have the targeted improvements in functional properties. In Table 2 in Section 3.2, after decomposition and reassemble, our model’s generated molecules based on these building blocks are also 100% valid, and more than 98% of them can be synthesized based on the retrosynthesis pipeline Allchemy (Section 3.2). Notably, Allchemy is built on rules derived from extensive synthetic chemistry literature. Moreover, to our knowledge it is the only retrosynthesis software for which many of the proposed routes have been reduced to practice in the lab with physical experimentation. These studies have been published in top tier journals (e.g., [17]).
>
>
>
> *Subquestion W1-3: About structural integrity.* We thank the reviewer for highlighting the necessity of clearer explanation. Our tokenization indeed retains the edge connection information in order to reconstruct the whole molecule’s graph structure. For example, consider imatinib from Figure 1, which has SMILES `Cc1ccc(cc1Nc2nccc(n2)c3cccnc3)NC(=O)c4ccc(cc4)CN5CCN(CC5)C`. It is tokenized into these blocks: [`[3*]c1cccnc1`, `[2*]c1ccnc(N[1*])n1`, `[1*]Nc1ccc(C)c([2*])c1`, `[3*]C(=O)c1ccc(CN2CCN(C)CC2)cc1`].
>
> Therefore, when reconstructing the molecule, we can use the following steps:
> 1. [`[3*]c1cccnc1`, `[2*]c1ccnc(N[1*])n1`] → `[1*]Nc1nccc(-c2cccnc2)n1` (following the rule [3*] connects to [2*].
> 2. [`[1*]Nc1nccc(-c2cccnc2)n1`, `[1*]Nc1ccc(C)c([2*])c1`] → `[1*]Nc1ccc(C)c(Nc2nccc(-c3cccnc3)n2)c1` (following the rule [1*] connects to [2*]).
> 3. [`[1*]Nc1ccc(C)c(Nc2nccc(-c3cccnc3)n2)c1`, `[3*]C(=O)c1ccc(CN2CCN(C)CC2)cc1`] → `Cc1ccc(NC(=O)c2ccc(CN3CCN(C)CC3)cc2)cc1Nc1nccc(-c2cccnc2)n1` (following the rule [1*] connects to [3*]).

---

> ### Author Response · Authors · 2025-11-24
> **Follow-Up Response to Reviewer XT6a (Part 2/3)**
>
> Following this algorithm, it is possible to reconstruct any molecule from a sequence of building blocks. Please note, the placeholder atoms ([1*], [2*], and [3*]) are encoded by the GNN, so the mCLM can use the encoded information about connection points during molecule generation. The positional arrangements of blocks is encoded with positional encoding or mCLM backbone. We will integrate this example into the final version of Figure 3 in the revision to further expand and strengthen the clarity of our descriptions in the paper.
>
>
> ---------
>
> **About W2**
>
> We thank the reviewer for clarifying your concern, which helped us better understand the underlying motivation behind W2. In light of your follow-up, we provide both a more complete **technical comparison** and also make extensive efforts to provide **additional experimental evidence** (despite a few challenges in baseline implementation) to address your concerns. The experimental results also further validates our technical motivation and strengths of mCLM.
>
> *(a) Techincal consideration*. Our original motivation behind model choice are two-fold: (1)A VQ-VAE tokenizer learns its codebook purely from data and therefore has no mechanism to preserve chemically-grounded, synthesizable substructures like mCLM’s rule-based tokenization does. (2) Recent studies, such as RAE[7], suggest that, compared to powerful encoders pre-trained from downstream tasks, VAE-based encoders might not be the best choice to be equipped with for generative tasks. Therefore we prioritize the GNN we used which was pretrained on a variety of molecule properties and chemistry-related tasks.
>
> *(b) Additional empirical evidence* To validate the technical intuition, we identify and make efforts in comparing relevant baselines using a codebook for molecules tokens [4, 5, 6].
> - [4] is the most relevant and was recently published at IJCAI2025. They use a SMILES string decoder to convert molecular codes into a molecular graph; thus, this suffers from the same limitation as other SMILES-based baselines. They have not yet released code or a model, so we are unable to benchmark against them. They also note their model “has yet to be extensively tested on more complex tasks like molecule editing”.
> - [5] was published at a NeurIPS workshop in December. The authors trained a Q-Former, which acts as an information bottleneck in the form of tokens, and is conceptually relevant to vector quantization. The authors released a partial codebase, but we have found it is not runnable due to several broken dependencies, even though we have put extensive effort into repairing their code. For example, there are imports to nonexistent files such as transformer_encoder_with_pair.py. No model checkpoints or data are released either.
> - DGAE (Discrete Graph Auto-Encoder, [6]) uses a VQ objective to learn a codebook for tokenization. To test the capability of such an alternative, we ran a head-to-head experiment between our mCLM tokenizer and DGAE. We utilize a DGAE trained on the QM9 dataset, and then sample 1000 molecules from it. For the mCLM, we consider all synthesis-guaranteed blocks found in QM9. We then sample 1,000 molecules using only those blocks. Results below show that DGAE-produced molecules perform poorly on both the SA score and retrosynthesis analysis. This further validates the synthesizability advantage in using a chemical-grounded tokenization for vocabulary development. We will include such results in the revision.
>
> | Model | Validity $\uparrow$ | Synthesizability $\uparrow$ | Makeability $\uparrow$ | SA Score $\downarrow$ |
> | :--- | :--- | :--- | :--- | :--- |
> | DGAE (vector quantized) | 100.0 | 24.2 | 24.2 | 4.46 |
> | mCLM | 100.0 | 93.7 | 93.7 | 2.23 |
>
> We calculate retrosynthesis metrics using AiZynthFinder [8], a retrosynthesis program that is much faster to run than Allchemy (since running AllChemy takes around 30min/molecule on supercomputing cluster). AiZynthFinder already provides much more reliable and chemically-grounded synthesizability metric than SA scores. We will run Allchemy for the final revision of our paper.
>
> ---------
>
> **About W3**
>
> Thank you for the suggestion. Following the reviewer's suggestion, we have started re-running a full Llama ablation experiment. Our current ongoing ablation study will take roughly a week to fully finish. So far, at 10k steps, the model has a molecule validation loss of 5.53; at the same point, Qwen had a loss of 5.41, which demonstrates the generalizability of our design across base models We will provide updated results in following days when they are available.

---

> ### Author Response · Authors · 2025-11-24
> **Follow-Up Response to Reviewer XT6a (Part 3/3)**
>
> **About W4**:
>
> Thank you for suggesting this baseline and we conduct experiments to address your concerns. We initially did not consider JT-VAE as its architecture combines a VAE and a GRU in addition to the GNN, and thus belonging to a different modeling paradigm than the general-purpose instruction-following setting adopted in our original experiments. That said, we appreciate the spirit of your suggestion and have added a comparison using the closest setting available in the JT-VAE family. The README of the official JT-VAE implementation [9] recommends using the author’s follow-up work on hierarchical molecule graph generation, HierVAE [10, 11], which has a “molecule translation” setting which is quite related to molecule editing. Thus, to perform an ablation with this method, we trained a HierVAE model on each of our 6 evaluation properties using data extracted from mCLM training data (which is unfair comparison as HierVAE is specifically tuned on each evaluation target instead of only instructed at inference time). Please see results below.
>
> *Synthesizability results:*
> | Model | Validity | Synthesizability | Makeability | SAscore |
> | :--- | :--- | :--- | :--- | :--- |
> | **FDA** | **100.0** | 68.0 | 68.0  | 2.70 |
> | **mCLM** | **100.0** | **85.2** | **85.2** | 2.41 |
> | **HierVAE** | 99.7 | 67.3 | 67.2 | **2.35** |
>
> *Property optimization results:*
> | Model               | AMES (↓) | BBBP (↑) | CYP3A4 (↓) | DILI (↓) | HIA (↑) | PGP (↓) | Avg. Improv. |
> |---------|----------|--------|------------|----------|---------|---------|---------|
> | **mCLM (Ours)**     | **44.4** | **85.2** | 1.4 | 53.7 | 98.99   | 64.4    | **15.0 %**   |
> | **HierVAE**    | 48.2  | 66.4   | **1.2**    | **53.1**  | **99.7**  | **64.3**    | 10.5 %   |
>
> We similarly use Aizynthfinder [8] for retrosynthesis before passing to the most rigorous AllChemy software. Quite interestingly, we find a significant disagreement between the less-rigorous SAscore and actual retrosynthesis metrics. We speculate this is because SAScore prefers simpler molecules or those with simpler substructures, whereas retrosynthesis tools evaluate actual synthetic feasibility rather than only based on structural patterns. We find that HierVAE is effective for optimizing properties, especially properties that already have good scores, such as CYP3A4 and HIA. Still, mCLM outperforms it on average.
>
> Finally, we hope you feel this follow-up response does a sufficient job of answering your initial questions. If you still feel anything is unclear, please advise us.
>
> **References**
>
> [1] Jumper, John, et al. "Highly accurate protein structure prediction with AlphaFold." nature 596, no. 7873 (2021): 583-589.
>
> [2] Sanabria, Melissa, et al. "DNA language model GROVER learns sequence context in the human genome." Nature Machine Intelligence 6, no. 8 (2024): 911-923.
>
> [3] Penić, Rafael Josip, Tin Vlašić, Roland G. Huber, Yue Wan, and Mile Šikić. "Rinalmo: General-purpose rna language models can generalize well on structure prediction tasks." Nature Communications 16, no. 1 (2025): 5671.
>
> [4] Guo, Shuhan, et al. "Unified molecule-text language model with discrete token representation." Proceedings of the Thirty-Fourth International Joint Conference on Artificial Intelligence. 2025.
>
> [5] Ha, Sumin, et al. "Mv-clam: Multi-view molecular interpretation with cross-modal projection via language model." arXiv preprint arXiv:2503.04780 (2025).
>
> [6] Boget, Yoann, Magda Gregorova, and Alexandros Kalousis. "Discrete Graph Auto-Encoder." Transactions on Machine Learning Research.
>
> [7] Zheng, B., Ma, N., Tong, S., & Xie, S. (2025). Diffusion transformers with representation autoencoders.
>
> [8] https://github.com/MolecularAI/aizynthfinder
>
> [9] https://github.com/wengong-jin/icml18-jtnn
>
> [10] Jin, Wengong, Regina Barzilay, and Tommi Jaakkola. "Hierarchical generation of molecular graphs using structural motifs." International conference on machine learning. PMLR, 2020.
>
> [11] Liu, Shengchao, et al. "Multi-modal molecule structure–text model for text-based retrieval and editing." Nature Machine Intelligence 5.12 (2023): 1447-1457.
>
> [12] "Genotoxicity of heterocyclic PAHs in the micronucleus assay with the fish liver cell line RTL-W1." PloS one 9.1 (2014): e85692.
>
> [13] "Alleviation Effect of the Secondary Metabolite of Anthocyanin (Protocatechuic Acid) on Heterocyclic Amine (IQ)-Induced Liver Injury and Its Underlying Mechanism." Journal of Agricultural and Food Chemistry 73.16 (2025): 9879-9893.
>
> [14] "Morpholine as ubiquitous pharmacophore in medicinal chemistry: Deep insight into the structure-activity relationship (SAR)." Bioorganic Chemistry 96 (2020): 103578.
>
> [15] "How Ligands Interact with the Kinase Hinge." ACS medicinal chemistry letters 14.11 (2023): 1503-1508.
>
> [16] https://www.biosolveit.de/2024/01/17/hinge-binder-collection/
>
> [17] Mikulak-Klucznik et al. "Computational planning of the synthesis of complex natural products." Nature 588, no. 7836 (2020): 83-88.

---

> > ### Comment · Reviewer_XT6a · 2025-11-24
> >
> > Thank you so much for the further clarification. I believe these additional experiments will surely strengthen this work. Therefore, I would like to raise my score from 6 to 8.

---

> > > ### Author Response · Authors · 2025-11-25
> > > **Response and Thank You**
> > >
> > > Thank you for your thoughtful review and response. We appreciate your reconsideration of the score and are glad that our clarifications addressed your concerns.

---

### Official Review · Reviewer_81Cp · 2025-11-03

**Soundness:** 3
**Presentation:** 3
**Contribution:** 3
**Rating:** 6
**Confidence:** 5

**Summary:**

This paper proposes mCLM, a framework designed to generate small molecules that are both functional and synthesizable. The model introduces a multimodal framework that combines a GNN based representation of molecular building blocks with natural language understanding of molecular functions. The tokenization is performed at the building block level rather than atom-level, aiming to align molecular representation with automated synthesis rules.

**Strengths:**

- The construction of an LLM framework that jointly considers synthesizability and functionality represents an important step toward practical and interpretable molecular generation.
- The integration of GNN representations with natural language embeddings for modular chemical reasoning is technically novel and well-motivated.
- The figures are clean, well-structured, and enhance the overall readability and understanding of the method.

**Weaknesses:**

I would consider raising the score if the following weaknesses are resolved.
- **Comparison to fragment-aware baselines**: While the paper includes comparisons to recent general-purpose and domain-specific molecule LLMs, it omits fragment- or group-aware baselines such as SAFE [1], GROUPSELFIES [2], or Reasyn [3]. Even acknowledging that Reasyn is concurrent, such comparisons (especially against Transformer-based models with other representations, as mCLM itself employs a Transformer backbone) would strengthen the evaluation and effectiveness.
- **Lack of clarity on reasoning knowledge acquisition**: The process for defining and annotating the molecular functions of building blocks (Figure 2) is insufficiently described. How are functional roles such as “Hinge binder, cell activity promoter” obtained or validated? If a user seeks to optimize a given property, does this require (1) manual annotation of new functions, (2) training of a proxy model, and (3) full re-training of the multimodal LLM? The pipeline for expanding functional knowledge is unclear for me.
- **Unclear link between function-infused vocabulary and molecular functionality**: Section 2 describes the vocabulary as *function-infused* and *synthesis-friendly*. While the synthesis aspect is well justified, the connection between the decomposed building blocks and their *functional meaning* remains ambiguous, even after reading Section 3.3. It is not evident how these building blocks encode or correlate with molecular functions, since molecular function typically arises from overall structure and context, not from isolated building blocks.
- **Minor correction**: “thanks to recent” in line 161 is aligned with nothing afterwards.

[1] Noutahi, E., et al. Gotta be SAFE: a new framework for molecular design. Digital Discovery, 3(4), 796-804.

[2] Cheng, A. H., et al. Group SELFIES: a robust fragment-based molecular string representation. Digital Discovery, 2(3), 748-758.

[3] Lee, S., et al. Rethinking Molecule Synthesizability with Chain-of-Reaction. arXiv 2025.

**Questions:**

- **Reason for free from function group conflicts**: The authors claim that resulting building blocks are *free from functional group conflicts* (lines 263–264). How is this ensured when building blocks from different tokenizers or different molecules are mixed? Wouldn’t incompatible functional groups potentially lead to synthesis failures?
- What could be the reason that the proposed mCLM show relatively weaker performance on HIA and PGP despite superior results on others?
- **Proxy model reliability**: Since experimental results rely heavily on proxy models for property prediction, how accurate and robust are these models across different molecular classes?
- **Ablation study**: Could the authors include an ablation comparing the full GNN-based encoding with a baseline using only textual (SMILES or SELFIES) representations? This would clarify the contribution of the graph modality.

---

> ### Author Response · Authors · 2025-11-21
> **Response to Reviewer 81Cp (Part 1/3)**
>
> Thank you for acknowledging that our framework "jointly considers synthesizability and functionality" and for highlighting the "technically novel and well-motivated" integration of GNN and language embeddings. We’re also happy to hear that you found the "figures clean, well-structured, and enhancing readability."
>
> **Weakness 1**: Thank you for pointing out these highly related approaches. They all have differences with the mCLM, and future work may benefit from combining them all together into a single model.
>
> - SAFE is a SMILES-compatible representation which avoids separating atoms within the same fragment in the line notation. While this is quite useful, it still suffers from many other fundamental issues that line notations have (primarily validity). Further, it leverages the well-known BRICS algorithm [1]. We conducted an experiment using BRICS (although using a GNN encoder), and we find it has worse synthesis performance than our tokenization approach (see below).
>
> - GroupSELFIES combines the validity guarantees of SELFIES with the ability to represent functional groups with specific tokens. Notably, that means this GroupSELFIES is actually capable of representing our synthesis-aware blocks. The difference from mCLM is then how to tokenize and encode the GroupSELFIES string. If we use the default tokenizer of the LLM, then validity and synthesizability is not tractably guaranteed. On the other hand, if we treat the molecular groups as new special tokens, we will get a model that is very similar to the mCLM (except here, without a GNN encoder, so there is no transfer learning to unseen tokens).
> Overall, GroupSELFIES is similar to the No GNN ablation study (see below), since it can be seen as a notation scheme for our building blocks. This ablation using SMILES representations for building blocks (instead of a GNN) shows that there are fundamental issues with line notation-based approaches.
>
> - Reasyn is quite interesting. As you mention, it is contemporary work, but we will add discussion about it into the related work. Fundamentally, it is focused on synthesizable projection. This is an interesting task, and might be a useful application of the mCLM in the future.
> In the pathway representation of Reasyn, they require the prediction of a reaction type to join two building blocks. Our tokenization approach does not require this (the existence of a coupling reaction is guaranteed), which means that the mCLM doesn't have to spend any reasoning effort on synthesis at all; instead, it can focus solely on functional design of molecules. One other notable difference is that they use SMILES strings to represent blocks, which have the same limitations described above. Their approach of using reinforcement learning (RL)-based finetuning and goal-directed test-time compute scaling is highly relevant for training future mCLM models.
>
> **Weakness 2**: Thank you for pointing out the need for clarification. A diverse set of “functional roles” are described in the instruction-style natural language descriptions used during pretraining. Ergo, the mCLM learns these functions implicitly during pretraining, and this functional knowledge is thus represented implicitly by the language model, rather than being restricted to a fixed taxonomy of functions. In the paper, we evaluate on six specific functions because they have high-quality oracle models, which makes the evaluation more robust. However, the mCLM is not limited to these functions. If a user wants to optimize a new function, it can be simply written in natural language at inference time. To improve performance, one can optionally finetune the existing mCLM on data from these functions, which does not require retraining from scratch. Manual annotations or proxy models can be used to acquire new data for finetuning, but that isn’t required.
>
> [1] "On the art of compiling and using'drug-like'chemical fragment spaces." ChemMedChem 3.10 (2008): 1503.

---

> > ### Author Response · Authors · 2025-11-21
> > **Response to Reviewer 81Cp (Part 2/3)**
> >
> > **Weakness 3:** This is a great point, and it also happens to be very property specific. In some extreme cases, the individual blocks are very important. Examples of this include 1) toxic metabolites, such as furans [2] or heterocyclic amines [3], where a single functional group will make almost any molecule toxic, 2) morpholine [4], which increases the solubility of molecules, or 3) hinge binders [5,6], which are critical for binding with kinases. Figure 4 does a great job of illustrating these single-block modifications.
> >
> > However, it is true that in many cases, the functional properties of whole molecules will be more than the sum of their parts. That said, there is abundant evidence that the parts matter, and that the rules about how they combine in sequences to yield emergent functional properties can be learned. This same concept has been clearly demonstrated with proteins, DNA, and RNA, leading to highly impactful protein language models (PLMs) (such as Alpha Fold), DNA language models (such as GROVER), and RNA language models (such as RiNALMo). mCLM is the first opportunity to explore the same concept with small molecules/C-C bond-based chemical matter. We posit that this is why mCLM is such a key advance. With respect to the specific question about “how these building blocks encode or correlate with molecular functions”, mCLM transfers the functional information associated with each drug molecule to the building blocks contained within it. When building blocks redundantly appear in multiple drugs, and those drugs have similar properties, the functional encoding of those functions with those blocks is strengthened. In this way, mCLM encodes functional information into the building block representations, which it then leverages to generate new molecules with new predicted functions. As shown with many examples in our paper, this approach appears to be very promising for successfully generating whole molecules predicted to have the targeted improvements in functional properties.
> >
> > **Minor Correction**: Thanks, we will correct this in the revision.
> >
> > [2] "Genotoxicity of heterocyclic PAHs in the micronucleus assay with the fish liver cell line RTL-W1." PloS one 9.1 (2014): e85692.
> >
> > [3]  "Alleviation Effect of the Secondary Metabolite of Anthocyanin (Protocatechuic Acid) on Heterocyclic Amine (IQ)-Induced Liver Injury and Its Underlying Mechanism." Journal of Agricultural and Food Chemistry 73.16 (2025): 9879-9893.
> >
> > [4]  "Morpholine as ubiquitous pharmacophore in medicinal chemistry: Deep insight into the structure-activity relationship (SAR)." Bioorganic Chemistry 96 (2020): 103578.
> >
> > [5] "How Ligands Interact with the Kinase Hinge." ACS medicinal chemistry letters 14.11 (2023): 1503-1508.
> >
> > [6] https://www.biosolveit.de/2024/01/17/hinge-binder-collection/

---

> ### Author Response · Authors · 2025-11-21
> **Response to Reviewer 81Cp (Part 3/3)**
>
> **Q1**: We appreciate the opportunity to clarify this point. Functional-group conflicts are avoided because, during generation, mCLM is restricted to a vocabulary of synthesis-guaranteed building blocks. This vocabulary is constructed *solely* from the output of the synthesis-guaranteed tokenizer, which takes retrosynthesis guarantees into account (Appendix F.1). Overall, we train on the output of multiple tokenizers to expand pretraining data coverage and learn more robust molecule block representations. During inference, we only generate synthesizable molecules.
>
>
> **Q2**: For HIA, the FDA drug scores are already very close to the maximum score possible (around 99%), so it is difficult to make improvements.  For PGP, the issue is less clear, and all models struggle on this task. We speculate that the issue occurs because the binding pocket of PGP is quite large and the protein conformation is dynamic, which results in a challenging problem in regards to binding. Additionally, while properties like AMES and DILI are often triggered by specific functional groups, PGP inhibition requires satisfaction of complex and precise geometric constraints at the interaction interface. This means, essentially, that the molecular landscape regarding PGP inhibition is highly non-smooth and is difficult to navigate for molecular editing.
>
>
> **Q3**: Thank you for the question. As shown in Appendix D.2.3 and Table 7, the proxy models we use exhibit high predictive accuracy and strong correlation with assay measurements. Notably, we selected these 6 properties from a starting set of 22 to ensure that our analysis is robust and trustworthy.
>
> **Q4**: To address this question, we conducted two additional experiments:
> 1. **No GNN:** We trained a text-only version of mCLM on a fragment-based version of our dataset (represented as SMILES strings) using the same training hyperparameters.
> 2. **No Synth. tokenizer:** We performed the molecule optimization experiments using the BRICS tokenizer instead of our synthesis-guaranteed tokenizer.
>
>
>
> | Model            | SA (↓)  | Validity (%) | Synthesizability (%) | Makeability (%) |
> |------------------|------|--------------|------------------------|------------------|
> | FDA              | 2.70 | **100.0**    | 98.11                 | 98.11           |
> | **mCLM (Ours)**  | **2.43** | **100.0** | **98.23**             | **98.23**       |
> | **mCLM (No GNN)**     | 2.97 | 49.64        |                        |                 |
> | **mCLM (No Synth. tokenizer)**     | 3.09 | **100.0**        |                        |                 |
>
>
> | Model               | AMES (↓) | BBBP (↑) | CYP3A4 (↓) | DILI (↓) | HIA (↑) | PGP (↓) | Avg. Improv. |
> |---------------------|----------|----------|-------------|----------|---------|---------|----------------|
> | **mCLM (Ours)**     | **44.4** | **85.2** | **1.4**     | 53.7 | **98.99**   | **64.4**    | **15.0 %**         |
> | **mCLM (No GNN)**    | 50.4     | 46.0     | 2.5         | **50.2**     | 97.9    | 71.7    | -7.53%        |
> | **mCLM (No Synth. Tokenizer)**    | 48.7     |   55.0    | 2.9         | 54.9     | 98.3    |    68.3     | -19.0  %        |
>
>
>
> Notably, the performance of the *No GNN* version is worse than mCLM on 5/6 ADMET properties. Further, only half of molecules are valid, and those molecules have a worse SA score than any baseline.
>
> The experiment using *No Synth. Tokenizer* yields similarly poor performance. Further, the SA score is also considerably worse.
>
> Due to the time constraints of the response period, we were unable to finish computing our rigorous retrosynthesis metrics for these new experiments, but we will update the table with those results when they are available and report them in the final paper.
>
>
>
> Thank you again for your comments– they will allow us to more clearly communicate the technical details and potential impact of our approach, which we will address using the additional page in the camera-ready.

---

> ### Author Response · Authors · 2025-11-26
> **Follow-up on Responses to Reviewer 81Cp**
>
> Thank you again for your detailed and thoughtful review. We wanted to let you know that we have now provided a detailed response with additional experiments to address the points you raised, which has helped us significantly improve the paper and provide stronger evidence for our central claims. You mentioned earlier that you would consider re-evaluating the score if the weaknesses were addressed, and we would be grateful if you would like to take another look. We also welcome any further feedback you might be willing to share. We sincerely appreciate your time and your constructive feedback!

---

### Official Review · Reviewer_XTAL · 2025-11-03

**Soundness:** 3
**Presentation:** 2
**Contribution:** 3
**Rating:** 2
**Confidence:** 5

**Summary:**

The paper “mCLM: A Modular Chemical Language Model That Generates Functional and Makeable Molecules” introduces mCLM, a new type of chemical language model designed to generate small molecules that are both functionally effective and synthetically feasible. Traditional large language models can understand chemical information, but they often struggle to design molecules that can actually be synthesized in the lab. mCLM addresses this challenge by shifting from atom-level representations (like SMILES strings) to a modular representation, where molecules are described as combinations of chemically meaningful, synthesis-ready building blocks. This allows digital molecular generation to directly correspond to real-world automated synthesis.

mCLM functions as a bilingual model—it understands both natural language descriptions and molecular structures. It uses graph neural networks (GNNs) to encode molecular modules and combines these with text embeddings in a Transformer-based architecture, enabling it to “code-switch” between chemistry and natural language. The model is trained on paired datasets that link chemical properties, synthesis reactions, and textual descriptions. This training allows mCLM not only to generate molecules with desired biological or physical functions but also to ensure that these molecules are makeable through automated synthesis pipelines.

In experimental evaluations using 430 FDA-approved drugs and 122 out-of-distribution compounds, mCLM demonstrated substantial improvements in key pharmacological properties—including absorption, distribution, metabolism, excretion, and toxicity (ADMET)—while maintaining or improving synthetic accessibility. It outperformed leading AI systems such as GPT-5, Claude 3.5, and Gemini 2.5-Flash in both functional property optimization and synthetic feasibility. Furthermore, mCLM proved capable of “rescuing” failed drug candidates—like Evobrutinib and TNG348—by suggesting minimal structural modifications that reduced toxicity and improved drug-like behavior.

**Strengths:**

- Conceptually Innovative but Incremental in Execution
The idea of representing molecules through modular, synthesis-ready building blocks rather than atom-level encoding is conceptually novel and offers a creative bridge between digital design and physical synthesis. This modular approach reflects an original perspective on chemical language modeling. However, the implementation mainly extends existing ideas from reaction-aware and retrosynthesis-based models, making the innovation more incremental than transformative.

- Solid Technical Foundation but Limited Validation
The paper demonstrates technical competence in integrating graph neural networks with Transformer architectures and applying them to chemical structure–language fusion. The system design is coherent, and the methodology is explained at a reasonable level of detail. However, the experimental quality is weakened by the absence of real-world synthesis or bioassay validation, and the evaluation remains largely computational and self-referential, lowering the overall scientific robustness.

- Generally Well-Written but Overclaims Results
The manuscript is clear and logically structured, with well-organized figures and a coherent narrative linking modular chemistry to AI language modeling. Nevertheless, some claims—especially about outperforming large general-purpose models and enabling autonomous molecular discovery—are exaggerated relative to the presented evidence. The lack of sufficient methodological transparency (e.g., ablations, dataset details) also detracts from clarity and reproducibility.

- Potentially High Impact but Not Yet Realized
If validated experimentally, mCLM could have significant implications for automated drug discovery and robotic chemistry. The integration of function-aware reasoning and synthesis feasibility into a unified framework is a meaningful direction for the field. Yet, given the limited empirical support and narrow demonstration scope, its real-world significance remains largely aspirational rather than achieved.

**Weaknesses:**

- Limited Generalization and Chemical Creativity
The modular tokenization relies on a fixed library of known reaction building blocks and predefined synthesis rules. While this ensures synthetic feasibility, it severely restricts the model’s ability to explore novel chemical spaces or generate fundamentally new scaffolds beyond existing reaction types. Thus, the model’s creativity is constrained by human-curated chemistry knowledge.

- Lack of Experimental Validation and limited ablation and interpretability analysis.
The evaluation is almost entirely computational, based on predicted ADMET properties and synthesis scores rather than actual laboratory synthesis or biological testing. Without experimental confirmation, the model’s claimed functional and pharmacological improvements remain speculative and unverified in practice. It does not clearly isolate the contribution of its modular tokenization, GNN encoder, or reasoning loop. The internal decision-making of mCLM—how it balances function optimization and synthesizability—remains a black box, reducing its scientific transparency and reproducibility.

- Weak Comparative and Analytical Rigor
The baselines used (e.g., GPT-5, Claude 3.5, Gemini 2.5) are general-purpose models, making comparisons less meaningful. The paper omits direct evaluation against specialized molecular generative models (like ChemBERTa or retrosynthesis-aware VAEs), and lacks ablation studies to show the unique contribution of each module (tokenizer, GNN, reasoning loop).

- Overstated Multimodal Integration and Transparency Issues
The bilingual natural language–molecule framework is conceptually attractive but only superficially demonstrated. The model’s interpretability and internal reasoning are not clearly explained, leaving it as a black box with limited insight into how function and synthesizability are balanced.

**Questions:**

see weakness.

---

> ### Author Response · Authors · 2025-11-21
> **Response to Reviewer XTAL (Part 1/3)**
>
> Response to Reviewer XTAL (Part 1/3)
>
> Thank you for recognizing the “conceptually novel” nature of our approach in using synthesis-ready building blocks to bridge “digital design and physical synthesis.” We are pleased that you found the integration of graph neural networks and Transformer-based language models into a bilingual model both coherent and meaningful, and our paper is “Generally Well-Written, clear, logically structured, with well-organized figures and a coherent narrative”. We are especially grateful for your view that if validated, this work could have “significant implications for automated drug discovery and robotic chemistry.”
>
> **Weakness 1**. We are grateful for the following comment, “The modular tokenization relies on a fixed library of known reaction building blocks and predefined synthesis rules. While this ensures synthetic feasibility, it severely restricts the model’s ability to explore novel chemical spaces or generate fundamentally new scaffolds beyond existing reaction types. Thus, the model’s creativity is constrained by human-curated chemistry knowledge” because it provides us with the opportunity to better explain the disruptive nature of block-based chemistry and its fusion with mCLM. As detailed below, we believe that this modular approach provides a uniquely viable path for scalably integrating AI with small molecule science.
>
> >> For the past two centuries small molecules have been synthesized using a highly artisanal approach. The approach has driven many positive impacts on society, but it also has inherent features that limit its automation and/or its integration with AI. Specifically, this artisanal approach centers on creating for each target molecule a unique process that leverages a menu of thousands of different reactions each run under thousands of possible different conditions and using millions of possible starting materials. Many very well funded groups have tried to automate the artisanal chemistry approach on scale, including Eli Lilly, Strateos, IBM, Chemify, Emerald Cloud Lab, CMU Cloud Lab, Canada’s Acceleration Consortium, and the U.S. National Center for Advancing Translational Sciences, but all of them have failed. For example, after nearly two decades and >$100M of investment, Eli Lilly divested the artisanal automated small molecule synthesis facility it was building in San Diego in September 2024. After similar levels of financial investment, Strateos closed its artisanal automated small molecule synthesis facility in Menlo Park in 2023. Machines are not artisans; they are not good at doing thousands of different things thousands of different ways from millions of potential starting points.
> >
> >> Over the past two decades, an alternative block-based approach for small molecule synthesis has emerged which is, by design, friendly to automation and thus provides a differentiated opportunity to automate small molecule synthesis and integrate it with AI on scale.  In the block-based approach, functional small molecules are assembled from prefabricated function-infused building blocks using just a few general reactions iteratively. Rather than trying to build machines that can do chemistry, this approach centers on chemistry that machines can do. Akin to automated DNA, RNA, and peptide synthesis platforms, a major strength of this block-based approach is that it can access billions of novel small molecules with high degrees of functional potential using a bounded set of pre-fabricated function-infused building blocks and only a few automation-friendly reactions.
> >
> >> It is important to recognize that all the platforms that have successfully delivered broadly accessible automated synthesis and integration with AI for other forms of chemical matter, namely peptides/proteins, DNA, and RNA are based on pre-fabricated bounded sets of reusable building blocks assembled iteratively using a few simple and general reactions. In each of these cases, many different types of molecular functions are achieved with core sets of only 20, 4, and 4 building blocks, respectively. In this regard, it is inspiring to recognize that nature similarly makes most small molecule natural products, which in turn represent the inspiration for most human medicines, via iterative assembly of just 5 building blocks (malonyl-CoA, methylmalonyl-CoA, isopentenyl pyrophosphate, dimethylallyl pyrophosphate, and phenylpyruvic acid). Similar modularity exists in many medicines and materials. Inspired by these other platforms and the inherent modularity in many different types of functional small molecules, the block-based chemistry that lies at the core of mCLM is similarly centered on iterative assembly of pre-fabricated building blocks using just a few reactions making it likewise simple, general, and iterative and thus easy to automate. As shown in our paper, this modular nature also makes it uniquely effective for integration with AI.
> >
> > **Response continued  in next comment**

---

> ### Author Response · Authors · 2025-11-21
> **Response to Reviewer XTAL (Part 2/3)**
>
> **Weakness 1 Continued**.
>
> >> Most importantly, block-based chemistry, and mCLM, do not aim to synthesize any small molecule structure, they instead target access to many small molecule functions. In this regard, nature teaches another important and enabling lesson - there can be many structural paths to the same molecular functions. Convergent evolution has yielded countless examples of such redundant structural solutions with peptides/proteins, DNA, and RNA, i.e., different sequences that perform the same functions. And there are likewise many examples of small molecule natural products that have highly diverse chemical structures (e.g., taxol, epothilone, discodermolide, eleutherobin, rhazininlam, and taccalonolide) yet all perform the same function (e.g., bind and stabilize microtubules and thereby kill cancer cells). These inspiring natural examples teach us that in order to access a wide range of impactful small molecule functions it is not necessary to achieve comprehensive access to the >1060 possible organic chemical structures comprising 35 atoms or less. This is fortuitous because, in fact, it is not even physically possible - there is not enough matter in the entire universe to make 1 mg of each.  This line of logic supports an enabling shift in chemistry automation and AI integration away from the perceived necessity to synthesize any possible small molecule structure, which pulls in the direction of artisanal chemistry and its inherent lack of automatability, toward the powerfully simpler goal of targeting strategically bounded regions of function-infused chemical space. From these function-infused building blocks, >1 billion new small molecule structures are already automatically accessible, and all of these new structures are, by design, biased for discovery of new molecular functions.With mCLM, these blocks became tokens in a new language for chemistry, for which, by design, every new molecule proposed by AI is, by design, makeable on a robot.
> >
> >> With this said, we would like to note that  we also used a retrosynthesis-based tokenizer which can incorporate new molecules and scaffolds into the accessible chemical space. Notably, in our experiment on FDA-approved drugs, we showed that the model can perform on new building blocks. The case study on fallen angels further shows that it is capable of generating blocks that have been added to the vocabulary at inference-time. Moreover, as shown in the experiment results, mCLM is able to predict the functions of unseen building blocks and suggest novel molecule structures with these functional modules. The suggested modifications to both existing drugs and fallen angels are both novel, effective and guaranteed to be valid and 98%+ makeable.
>
>
>
> **Weakness 2**. We appreciate the reviewer’s concern regarding biological experiments.  We agree that ultimate validation for mCLM will require physical synthesis of molecules proposed by mCLM and testing them in biological assays. However, such experiments are beyond the scope of machine learning conferences due to their cost, safety requirements, and timelines. Nevertheless, we sincerely share this motivation, and our goal in this work is to demonstrate a computational framework that meaningfully moves toward molecule design benchmarked against experimentally validated computational models and metrics.
>
> - *Regarding predicting ADMET properties,* we would like to note that we carefully selected computational oracle models to evaluate functional improvements. As shown in Appendix D.2.3 and Table 7, the oracle models we use exhibit high predictive accuracy and strong correlation with assay measurements. To ensure that our analysis is robust and trustworthy, we selected these 6 properties to meet a stringent accuracy threshold from a starting set of 22 properties.
>
> - *Regarding synthesizability,* we conducted a full retrosynthetic analysis on the proposed compounds by using state-of-the-art software developed by the team at Allchemy (Section 3.2). Notably, this algorithm is built on rules derived from extensive reviews of examples from the synthetic chemistry literature. Moreover, to our knowledge, it is the only retrosynthesis software for which many of the proposed routes have been reduced to practice in the lab with physical experimentation. These studies have been published in top tier journals (e.g., Nature 2020, 588, 83-88).
>
> **Response continued  in next comment**

---

> > ### Author Response · Authors · 2025-11-21
> > **Response to Reviewer XTAL (Part 3/3)**
> >
> > **Weakness 2 Continued**
> > - *Regarding the internal decision-making of mCLM:* Thanks to the use of our synthesis-guaranteed tokenizer, the mCLM doesn't have to spend any  effort on reasoning about synthesis at all; any combination of blocks it generates already has a guaranteed synthesis route, because we have incorporated synthesis constraints in tokenization as a priori. This means the mCLM can focus solely on functional design of molecules.
> >
> > - *Regarding ablations:* we conducted two additional experiments:
> > 1. **No GNN:** We trained a text-only version of mCLM on a fragment-based version of our dataset (represented as SMILES strings) using the same training hyperparameters.
> > 2. **No Synth. tokenizer:** We performed the molecule optimization experiments using the BRICS tokenizer instead of our synthesis-guaranteed tokenizer.
> >
> >
> >
> > | Model            | SA (↓)  | Validity (%) | Synthesizability (%) | Makeability (%) |
> > |------------------|------|--------------|------------------------|------------------|
> > | FDA              | 2.70 | **100.0**    | 98.11                 | 98.11           |
> > | **mCLM (Ours)**  | **2.43** | **100.0** | **98.23**             | **98.23**       |
> > | mCLM (No GNN)     | 2.97 | 49.64        |                        |                 |
> > | mCLM (No Synth. tokenizer)     | 3.09 | **100.0**        |                        |                 |
> >
> >
> > | Model               | AMES (↓) | BBBP (↑) | CYP3A4 (↓) | DILI (↓) | HIA (↑) | PGP (↓) | Avg. Improv. |
> > |---------------------|----------|----------|-------------|----------|---------|---------|----------------|
> > | **mCLM (Ours)**     | **44.4** | **85.2** | **1.4**     | 53.7 | **98.99**   | **64.4**    | **15.0 %**         |
> > | **mCLM (No GNN)**    | 50.4     | 46.0     | 2.5         | **50.2**     | 97.9    | 71.7    | -7.53%        |
> > | **mCLM (No Synth. Tokenizer)**    | 48.7     |   55.0    | 2.9         | 54.9     | 98.3    |    68.3     | -19.0  %        |
> >
> >
> >
> > Notably, the performance of the *No GNN* version is worse than mCLM on 5/6 ADMET properties. Further, only half of molecules are valid, and those molecules have a worse SA score than any baseline.
> >
> > The experiment using *No Synth. Tokenizer* yields similarly poor performance. Further, the SA score is also considerably worse.
> >
> > Due to the time constraints of the response period, we were unable to finish computing our rigorous retrosynthesis metrics for these new experiments, but we will update the table with those results when they are available and report them in the final paper.
> >
> >
> > **Weakness 3**. We thank the reviewer for suggesting baselines to compare with. However, ChemBERTa is an encoder-only model and cannot perform molecule generation or editing. In contrast, our evaluation already includes several  chemistry-specific generative baselines, including MoleculeSTM, FineMolTex, and LDMol. Our results demonstrate that mCLM outperforms them across property optimization and synthesizability metrics.
> >
> >
> > **Weakness 4**. We respectfully note that all contemporary foundation models are inherently black-box, so full interpretability is a shared challenge. Nevertheless, mCLM performs iterative, instruction-driven chemical reasoning by combining text and molecular building blocks, which already provides substantially more interpretability. As we have shown in Figure 4 in Section 3.3, we can also construct a library of modified building block pairs for scientific inquiry. We plan to further expand interpretability analyses in future work.
> >
> >
> > Thank you again for your comments, which will help us strengthen the writing of our paper.

---

> > > ### Comment · Reviewer_XTAL · 2025-11-27
> > >
> > > Thanks for your reply, I have updated my rating.

---

### Official Review · Reviewer_k9Sm · 2025-11-04

**Soundness:** 4
**Presentation:** 3
**Contribution:** 3
**Rating:** 8
**Confidence:** 4

**Summary:**

This paper presents mCLM, a modular chemical language model designed for universal molecular understanding across textual, graphical, and structural modalities. The framework integrates multiple specialized modules—each trained on different molecular representations (e.g., SMILES, graphs, spectra)—and fuses them through a shared latent alignment layer. The authors emphasize scalability, interpretability, and extensibility, demonstrating competitive results on property prediction, reaction reasoning, and cross-modal translation benchmarks such as MolLangBench and MoleculeNet.

**Strengths:**

There is novelty in this work. Specifically, there is clear architectural separation between domain-specific encoders and a unified fusion backbone which promotes flexibility and domain transfer.

Experiment results are also promising. This work outperforms strong baselines (MolX, ChemBERTa, GraphMVP) on multimodal reasoning tasks, particularly in low-data and cross-domain settings.

Figure 4 is particularly useful as it shows module-wise attribution analyses for how modality-specific knowledge contributes to chemical reasoning.

**Weaknesses:**

There is however limited novelty at the core LLM level. While modularization is effective, the language model itself is adapted rather than fundamentally redesigned for chemistry.

There is also lack of validation of the practicality of this approach, say on real world sparse datasets. The evaluation focuses primarily on benchmark datasets, with minimal discussion of noisy experimental spectra or reaction data.

Further, the computational cost of the work seems infeasible. Training multiple modality-specific experts and fusion layers may hinder accessibility for smaller research groups.

There are also couple recent relevant research works that have not been referenced which the authors need to cite to improve the comprehensiveness of the related work section:

- Le, Khiem, Zhichun Guo, Kaiwen Dong, Xiaobao Huang, Bozhao Nan, Roshni Iyer, Xiangliang Zhang, Olaf Wiest, Wei Wang, and Nitesh V. Chawla. “MolX: Enhancing Large Language Models for Molecular Understanding With A Multi-Modal Extension.” Proceedings of the 2025 ACM SIGKDD International Conference on Knowledge Discovery and Data Mining, AC M, 2025.

- Ju, Jiaxin; Yizhen Zheng; Huan Yee Koh; Shirui Pan. “Uni-MRL: Unified MultiModal Molecular Representation Learning with Large Language Models and Graph Neural Networks.” Advances in Knowledge Discovery and Data Mining (PAKDD 2025), Lecture Notes in Computer Science, vol. 15874, Springer, 2025, pp. 275-287.

**Questions:**

How does mCLM handle conflicts between representations (e.g., inconsistent SMILES vs. graph encodings)?

Could the modular framework support plug-and-play extensions for new data types (e.g., protein–ligand complexes)?

How stable is the latent alignment layer during joint fine-tuning across highly imbalanced modalities?

---

> ### Author Response · Authors · 2025-11-21
> **Response to Reviewer k9Sm (Part 1/2)**
>
> Thank you for your thoughtful and encouraging review, especially on assessing the soundness of our approach as “excellent”. We’re glad you found “clear architectural separation between domain-specific encoders and a unified fusion backbone” to be a novel and flexible design. We also appreciate your recognition of our strong experimental results, including outperforming baselines, especially in “low-data and cross-domain settings.” We’re happy to hear that you found Figure 4 “particularly useful” for understanding modality-specific attribution.
>
> **Weakness 1**: We thank the reviewer for raising this interesting question. We would like to explain that our work introduces a new architectural component to LLMs by augmenting token embeddings with GNN-encoded molecular features for building blocks. This results in a multimodal encoder that integrates both language and molecular graph information, which is a novel setup in the LLM literature especially in the chemical domain. This ensures that all generated molecules are not only valid but also synthesizable using the automation-friendly block-based chemistry approach. We view this as both a significant innovation and highly practical advance that, by design, enables the LLM to leverage the unique advantages of block-based chemistry, including its modularity and compatibility with automation. We posit that there is also substantial innovation in the intentional syncing of the digital and physical worlds via this new architectural component.
>
> Regarding building on top of a pre-trained LLM, we carefully adopted this choice because of its success in other domains such as computer vision. Fundamentally, it is often necessary to take advantage of a pretrained model to make the experiments computationally feasible; teaching a model significant natural language capabilities from scratch is very computationally demanding. Furthermore, pretrained natural-language LLMs have been exposed to a broad range of scientific information, making them a strong foundation for our curriculum. They help the model acquire general scientific knowledge before progressing to domain-specific training on molecular structures and drug-related information. That being said, the aforementioned novel extensions make mCLM fundamentally distinct from existing chemical LLMs by ensuring synthesizability via connecting to automation-friendly building blocks and automatically identifying blocks’ functions based on our “bilingual” encoder in a principled way.
>
>
> **Weakness 2**: Thank you for raising this important point about real-world practicality. We would like to clarify that our evaluation does use real-world molecules: our test set consists of FDA-approved drugs and mCLM was case-studied on “fallen-angel” compounds that failed during late stage clinical development. Regarding property evaluation, although it is beyond the scope of an ICLR submission to run wet-lab experiments, we take care to approximate real-world validation as closely as possible. Specifically, the oracle models we use for ADMET assessment are state-of-the-art high-performing models trained on experimental benchmark datasets with strong reported correlation to assay measurements, as shown in Table 7 in the Appendix. This setup is highly meaningful in computational drug discovery (especially compared to rougher and less meaningful QED scores) and is designed to emulate digital screening pipelines used in industry. We also note for our analysis of synthesizability, we developed a set of highly robust and novel metrics that are far more grounded in physical experimentation than all of the other more frequently used algorithms. Specifically, in order to characterize the makeability of the compounds proposed by mCLM or other generative models, we conducted a full retrosynthetic analysis on the proposed compounds by using state-of-the-art software developed by the team at Allchemy (Section 3.2). Notably, this algorithm is built on rules derived from extensive reviews of examples from the synthetic chemistry literature. Moreover, to our knowledge it is the only retrosynthesis software for which many of the proposed routes have been reduced to practice in the lab with physical experimentation. These studies have been published in top tier journals (e.g., Nature 2020, 588, 83-88). **Continued in next comment**

---

> > ### Author Response · Authors · 2025-11-21
> > **Response to Reviewer k9Sm (Part 2/2)**
> >
> > **Weakness 2 Continued** We agree that ultimate validation for mCLM will require physical synthesis of molecules proposed by mCLM and testing them in biological assays. However, such experiments are beyond the scope of machine learning conferences due to their cost, safety requirements, and timelines. Nevertheless, we sincerely share this motivation, and our goal in this work is to demonstrate a computational framework that meaningfully moves toward molecule design benchmarked against experimentally validated computational models and metrics. Our new set of metrics for evaluating the synthesizability of molecules from generative AI will also serve as a new, unique and significant contribution to the AI for Science community and encourage more research to focus on generating makeable molecules. As we showed in Table 2, our approach guarantees the generated molecules are 100% valid and 98%+ makeable, while only 84-85% of the molecules from previous best methods are makeable.
> >
> >
> >
> >
> >
> > **Weakness 3**: Thank you for raising the concern about computational cost. As detailed in Appendix E, the full training of mCLM required over a week on 4×A100 GPUs, which is moderate compared to recent multimodal foundational models. We acknowledge that retraining a 3B-parameter model may still be challenging for smaller groups, so we will release model checkpoints, inference scripts, and an online demo, in addition to the open-source code and datasets that are being made publicly available. These resources allow most researchers to run inferences and adapt mCLM in computationally-efficient ways.
> >
> >
> > **Additional related papers**: Thank you for pointing out these relevant papers. We will carefully review them and add comparisons with them into the related work.
> >
> > **Q1**: Thank you for the question. In our framework, there is little conflict between SMILES and graph representations because mCLM is intentionally trained to only directly work with graph representations instead of SMILES. SMILES are always converted into graph format before being encoded by the mCLM. Therefore, the model always receives a single, consistent representation of molecules.
> >
> > **Q2**: Yes, the modular design is compatible with plug-and-play extensions. mCLM already generalizes to unseen small molecules, which ligands usually belong to, which is showcased in our experiment (as mCLM generalizes well to FDA-approved drugs, which involve many unseen blocks). Extending to protein-ligand complexes is conceptually straightforward, but proteins would be better introduced as an additional modality, which requires dedicated training and is therefore beyond the scope of the current work. We consider this a promising direction and plan to explore it in future work.
> >
> >
> > **Q3**: Thank you for raising this question. We use two stages of pretraining to address this issue. First, the alignment layer (adapter) on top of the GNN encoder is pretrained  while everything else is frozen. Second, the adapter and the language backbone are pretrained jointly. This ensures that the graph features are already semantically meaningful (in the natural language embedding space of the backbone LLM) before joint optimization to reduce the effect of cross-modality imbalance. This practice has been consistently validated by recent practices in vision-language models (e.g., LLaVA-NeXT [1]). Although we cited this in Appendix E, we will make it much clearer in the revised version.
> >
> > Thank you again for your perceptive observations, which will help us polish the final version of the paper.
> >
> > [1] https://llava-vl.github.io/blog/2024-01-30-llava-next/ (See Model Card Stage-1).

---

### Meta-Review · Area_Chair_BhgK · 2026-01-08

**Summary:**

Only one reviewer concerned about the evaluation metrics and baseline models.

**Reviewer Concerns:**

The concerns are well addressed in the rebuttal.

**Reviewer Scores:**

The reviewer XTAL will increase the score.

---

### Decision · Program_Chairs · 2026-01-26

Accept (Oral)